# Sparse High-Dimensional Isotonic Regression

**David Gamarnik** [*]
Sloan School of Management
Massachusetts Institute of Technology
Cambridge, MA 02139
gamarnik@mit.edu

**Julia Gaudio**[†]
Operations Research Center
Massachusetts Institute of Technology
Cambridge, MA 02139
jgaudio@mit.edu

## Abstract

We consider the problem of estimating an unknown coordinate-wise monotone function given noisy measurements, known as the isotonic regression problem. Often, only a small subset of the features affects the output. This motivates the *sparse* isotonic regression setting, which we consider here. We provide an upper bound on the expected VC entropy of the space of sparse coordinate-wise monotone functions, and identify the regime of statistical consistency of our estimator. We also propose a linear program to recover the active coordinates, and provide theoretical recovery guarantees. We close with experiments on cancer classification, and show that our method significantly outperforms several standard methods.

## 1 Introduction

Given a partial order $\preceq$ on $\mathbb{R}^d$, we say that a function $f : \mathbb{R}^d \to \mathbb{R}$ is *monotone* if for all $x_1, x_2 \in \mathbb{R}^d$ such that $x_1 \preceq x_2$, it holds that $f(x_1) \leq f(x_2)$. In this paper, we study the univariate isotonic regression problem under the standard Euclidean partial order. Namely, we define the partial order $\preceq$ on $\mathbb{R}^d$ as follows: $x_1 \preceq x_2$ if $x_{1,i} \leq x_{2,i}$ for all $i \in \{1, \ldots, d\}$. If $f$ is monotone according to the Euclidean partial order, we say $f$ is *coordinate-wise monotone*.

This paper introduces the *sparse isotonic regression problem*, defined as follows. Write $x_1 \preceq_A x_2$ if $x_{1,i} \leq x_{2,i}$ for all $i \in A$. We say that a function $f$ on $\mathbb{R}^d$ is *s-sparse coordinate-wise monotone* if for some set $A \subseteq [d]$ with $|A| = s$, it holds that $x_1 \preceq_A x_2 \implies f(x_1) \leq f(x_2)$. We call $A$ the *set of active coordinates*. The sparse isotonic regression problem is to estimate the $s$-sparse coordinate-wise *monotone* function $f$ from samples, knowing the sparsity level $s$ but not the set $A$. Observe that if $x$ and $y$ are such that $x_i = y_i$ for all $i \in A$, then $x \preceq_A y$ and $y \preceq_A x$, so that $f(x) = f(y)$. In other words, the value of $f$ is determined by the active coordinates.

We consider two different noise models. In the Noisy Output Model, the input $X$ is a random variable supported on $[0, 1]^d$, and $W$ is zero-mean noise that is independent from $X$. The model is $Y = f(X) + W$. Let $\mathcal{R}$ be the range of $f$ and let supp$(W)$ be the support of $W$. We assume that both $\mathcal{R}$ and supp$(W)$ are bounded. Without loss of generality, let $\mathcal{R} + $ supp$(W) \subseteq [0, 1]$, where $+$ is the Cartesian sum. In the Noisy Input Model, $Y = f(X + W)$, and we exclusively consider the classification problem, namely $f : \mathbb{R}^d \to \{0, 1\}$. In either noise model, we assume that $n$ independent samples $(X_1, Y_1), \ldots, (X_n, Y_n)$ are given.

The goal of our paper is to produce an estimator $\hat{f}_n$ and give statistical guarantees for it. To our knowledge, the only work that provides statistical guarantees on isotonic regression estimators in the Euclidean partial order setting with $d \geq 3$ is the paper of Han et al ([13]). The authors give

---

[*]http://web.mit.edu/gamarnik/www/home.html

[†]http://web.mit.edu/jgaudio/www/index.html

guarantees of the empirical $L_2$ loss, defined as $R(\hat{f}_n, f) = \mathbb{E}\left[\frac{1}{n}\sum_{i=1}^{n}\left(\hat{f}_n(X_i) - f(X_i)\right)^2\right]$, where the expectation is over the samples $X_1, \ldots X_n$. In this paper, we instead expand on the approach in Gamarnik ([11]), to the high-dimensional sparse setting. It is shown in [11] that the expected Vapnik-Chervonenkis entropy of the class of coordinate-wise monotone functions grows subexponentially. The main result of [11] is a guarantee on the tail of $\|\hat{f}_n - f\|_2$. When $X \in [0,1]^2$ and $Y \in [0,1]$ almost surely, it is claimed that

$$\mathbb{P}\left(\|\hat{f}_n - f\|_2 > \epsilon\right) \leq e^{\left\lceil\frac{4}{\epsilon^2}\right\rceil\sqrt{n} - \frac{\epsilon^4 n}{256}},$$

where $\hat{f}_n$ is a coordinate-wise monotone function, estimated based on empirical mean squared error. However, the constants of the result are incorrect due to a calculation error, which we correct. This result shows that the estimated function converges to the true function in $L_2$, almost surely ([11]). In this paper, we extend the work of [11] to the sparse high-dimensional setting, where the problem dimension $d$ and the sparsity $s$ may diverge to infinity as the sample size $n$ goes to infinity.

We propose two algorithms for the estimation of the unknown $s$-sparse coordinate-wise monotone function $f$. The *simultaneous* algorithm determines the active coordinates and the estimated function values in a single optimization formulation based on integer programming. The *two-stage* algorithm first determines the active coordinates via a linear program, and then estimates function values. The sparsity level is treated as constant or moderately growing. We give statistical consistency and support recovery guarantees for the Noisy Output Model, analyzing both the simultaneous and two-stage algorithms. We show that when $n = \max\left\{e^{\omega(s^2)}, \omega\left(s\log d\right)\right\}$, the estimator $\hat{f}_n$ from the simultaneous procedure is statistically consistent. In particular, when the sparsity $s$ level is of constant order, the dimension $d$ is allowed to be much larger than the sample size. We note that, remarkably, when the maximum is dominated by $\omega(s\log d)$, our sample performance nearly matches the one of high-dimensional linear regression [2, 10]. For the two-stage approach, we show that if a certain signal strength condition holds and $n = \max\left\{se^{\omega(s^2)}, \omega(s^3\log d))\right\}$, the estimator is consistent. We also give statistical consistency guarantees for the simultaneous and two-stage algorithms in the Noisy Input Model, assuming that the components of $W$ are independent. We show that in the regime where a signal strength condition holds, $s$ is of constant order, and $n = \omega(\log d)$, the estimators from both algorithms are consistent.

The isotonic regression problem has a long history in the statistics literature; see for example the books [19] and [20]. The emphasis of most research in the area of isotonic regression has been the design of algorithms: for example, the Pool Adjacent Violators algorithm ([15]), active set methods ([1], [5]), and the Isotonic Recursive Partitioning algorithm ([16]). In addition to the univariate setting ($f : \mathbb{R}^d \to \mathbb{R}$), the multivariate setting ($f : \mathbb{R}^d \to \mathbb{R}^q$, $q \geq 2$) has also been considered; see e.g. [21] and [22]. In the multivariate setting, whenever $x_1 \preceq x_2$ according to some defined partial order $\preceq$, it holds that $f(x_1)\tilde{\preceq}f(x_2)$, where $\tilde{\preceq}$ is some other defined partial order. There are many applications for the coordinate-wise isotonic regression problem. For example, Dykstra and Robertson (1982) showed that isotonic regression could be used to predict college GPA from standardized test scores and high school GPA. Luss et al (2012) applied isotonic regression to the prediction of baseball players' salaries, from the number of runs batted in and the number of hits. Isotonic regression has found rich applications in biology and medicine, particularly to build disease models ([16], [23]).

The rest of the paper is structured as follows. Section 2 gives the simultaneous and two-stage algorithms for sparse isotonic regression. Section 3 and Section A of the supplementary material provide statistical consistency and recovery guarantees for the Noisy Output and Noisy Input models. All proofs can be found in the supplementary material. In Section 4, we provide experimental evidence for the applicability of our algorithms. We test our algorithm on a cancer classification task, using gene expression data. Our algorithm achieves a success rate of about $96\%$ on this task, significantly outperforming the $k$-Nearest Neighbors classifier and the Support Vector Machine.

## 2    Algorithms for sparse isotonic regression

In this section, we present our two algorithmic approaches for sparse isotonic regression: the simultaneous and two-stage algorithms. Recall that $\mathcal{R}$ is the range of $f$. In the Noisy Output Model, $\mathcal{R} \subseteq [0,1]$, and in the Noisy Input Model, $\mathcal{R} = \{0,1\}$. We assume the following throughout.

**Assumption 1.** *For each $i \in A$, the function $f(x)$ is not constant with respect to $x_i$, i.e.*

$$\int_{x \in \mathcal{X}} \left| f(x) - \int_{z \in \mathcal{X}} f(z)dz \right| dx > 0.$$

## 2.1 The Simultaneous Algorithm

The simultaneous algorithm solves the following problem.

$$\min_{A,F} \sum_{i=1}^{n} (Y_i - F_i)^2 \tag{1}$$

$$\text{s.t. } |A| = s \tag{2}$$

$$F_i \leq F_j \qquad\qquad \text{if } X_i \preceq_A X_j \tag{3}$$

$$F_i \in \mathcal{R} \qquad\qquad \forall i \tag{4}$$

The estimated function $\hat{f}_n$ is determined by interpolating from the pairs $(X_1, F_1), \ldots, (X_n, F_n)$ in a straightforward way. In particular, $\hat{f}_n(x) = \max\{F_i : X_i \preceq x\}$. In other words, we identify all points $X_i$ such that $X_i \preceq x$ and select the smallest consistent function value. We call this the "min" interpolation rule because it selects the smallest interpolation values. The "max" interpolation rule is $\hat{f}_n(x) = \min\{F_i : X_i \succeq x\}$.

**Definition 1.** *For inputs $X_1, \ldots, X_n$, let $q(i, j, k) = 1$ if $X_{i,k} > X_{j,k}$, and $q(i, j, k) = 0$ otherwise.*

Problem (1)-(4) can be encoded as a single mixed-integer convex minimization. We refer to the resulting Algorithm 1 as Integer Programming Isotonic Regression (IPIR) and provide its formulation below. Binary variables $v_k$ indicate the estimated active coordinates; $v_k = 1$ means that the optimization program has determined that coordinate $k$ is active. The variables $F_i$ represent the estimated function values at data points $X_i$.

---

**Algorithm 1** Integer Programming Isotonic Regression (IPIR)

---

**Input:** Values $(X_1, Y_1), \ldots, (X_n, Y_n)$; sparsity level $s$
**Output:** An estimated function $\hat{f}_n$
 1: Solve the following optimization problem.

$$\min_{v,F} \sum_{i=1}^{n} (Y_i - F_i)^2 \tag{5}$$

$$\text{s.t. } \sum_{k=1}^{d} v_k = s \tag{6}$$

$$\sum_{k=1}^{d} q(i, j, k) v_k \geq F_i - F_j \qquad\qquad \forall i, j \in \{1, \ldots, n\} \tag{7}$$

$$v_k \in \{0, 1\} \qquad\qquad \forall k \in \{1, \ldots, d\} \tag{8}$$

$$F_i \in \mathcal{R} \qquad\qquad \forall i \in \{1, \ldots, n\} \tag{9}$$

 2: Return the function $\hat{f}_n(x) = \max\{F_i : X_i \preceq x\}$.

---

We claim that Problem (5)-(9) is equivalent to Problem (1)-(4). Indeed, the monotonicity requirement is $X_i \preceq_A X_j \implies f(X_i) \leq f(X_j)$. The contrapositive of this statement is $f(X_i) > f(X_j) \implies X_i \not\preceq_A X_j$; alternatively, $f(X_i) > f(X_j) \implies \exists k \in A$ s.t. $X_{ik} > X_{jk}$. The contrapositive is expressed by Constraints (7).

Recall that in the Noisy Input Model, the function $f$ is binary-valued, i.e. $\mathcal{R} = \{0, 1\}$. Let $\mathcal{S}^+ = \{i : Y_i = 1\}$ and $\mathcal{S}^- = \{i : Y_i = 0\}$. When $\{F_i\}_{i=1}^n$ are binary-valued, it holds that $\sum_{i=1}^{n} (Y_i - F_i)^2 = \sum_{i \in \mathcal{S}^+} (1 - F_i) + \sum_{i \in \mathcal{S}^-} F_i$. Therefore, if we replace the objective function (5) by $\sum_{i \in \mathcal{S}^+} (1 - F_i) + \sum_{i \in \mathcal{S}^-} F_i$, we obtain an equivalent linear integer program.

Algorithm 1 when applied to the Noisy Output Model is a mixed-integer convex optimization program. When applied to the Noisy Input Model, it s a mixed integer linear optimization program. While both are formally NP-hard in general, moderately-sized instances are solvable in practice.

## 2.2 The Two-Stage Algorithm

Algorithm 1 is slow, both in theory and in practice. Motivated by this, we propose an alternative two-stage algorithm. The two-stage algorithm estimates the active coordinates through a linear program, using these to then estimate the function values. The process of estimating the active coordinates is referred to as *support recovery*. The active coordinates may be estimated all at once (Algorithm 2) or sequentially (Algorithm 3). Algorithm 2 is referred to as Linear Programming Support Recovery (LPSR) and Algorithm 3 is referred to as Sequential Linear Programming Support Recovery (S-LPSR). The two-stage algorithm for estimating $\hat{f}_n$ first estimates the set of active coordinates using the LPSR or S-LPSR algorithm, and then estimates the function values. The results algorithm is referred to as Two Stage Isotonic Regression (TSIR) (Algorithm 4).

---

**Algorithm 2** Linear Programming Support Recovery (LPSR)

---

**Input:** Values $(X_1, Y_1), \ldots, (X_n, Y_n)$; sparsity level $s$
**Output:** The estimated support, $\hat{A}$
 1: Solve the following optimization problem.

$$\min_{v,c} \sum_{i=1}^{n} \sum_{j=1}^{n} \sum_{k=1}^{d} c_k^{ij} \tag{10}$$

$$\text{s.t.} \ \sum_{k=1}^{d} v_k = s \tag{11}$$

$$\sum_{k=1}^{d} q(i,j,k)\left(v_k + c_k^{ij}\right) \geq 1 \qquad \text{if } Y_i > Y_j \text{ and } \sum_{k=1}^{d} q(i,j,k) \geq 1 \tag{12}$$

$$0 \leq v_k \leq 1 \qquad \forall k \in \{1, \ldots, d\} \tag{13}$$

$$c_k^{ij} \geq 0 \qquad \forall i \in \{1, \ldots, n\}, j \in \{1, \ldots, n\}, k \in \{1, \ldots, p\} \tag{14}$$

 2: Determine the $s$ largest values $v_i$, breaking ties arbitrarily. Let $\hat{A}$ be the set of the corresponding $s$ indices.

---

In Problem (10)-(14), the $v_k$ variables are meant to indicate the active coordinates, while the $c_k^{ij}$ variables act as correction in the monotonicity constraints. For example, if for one of the constraints (12), $\sum_{k=1}^{d} q(i,j,k)v_k = 0.7$, then we will need to set $c_k^{ij} = 0.3$ for some $(i,j,k)$ such that $q(i,j,k) = 1$. The $v_k$'s should therefore be chosen in a way to minimize the correction.

Algorithm 3 determines the active coordinates one at a time, setting $s = 1$ in Problem (10)-(14). Once a coordinate $i$ is included in the set of active coordinates, variable $v_i$ is set to zero in future iterations.

---

**Algorithm 3** Sequential Linear Programming Support Recovery (S-LPSR)

---

**Input:** Values $(X_1, Y_1), \ldots, (X_n, Y_n)$; sparsity level $s$
**Output:** The estimated support, $\hat{A}$
 1: $B \leftarrow \emptyset$
 2: **while** $|B| < s$ **do**
 3:     Solve the optimization problem in Algorithm 2 with $s = 1$:

$$\min \sum_{i=1}^{n} \sum_{j=1}^{n} \sum_{k=1}^{d} c_k^{ij} \tag{15}$$

$$\text{s.t.} \ \sum_{k=1}^{d} v_k = 1 \tag{16}$$

$$v_i = 0 \qquad \forall i \in B \tag{17}$$

$$\sum_{k=1}^{d} q(i,j,k)\left(v_k + c_k^{ij}\right) \geq 1 \qquad \text{if } Y_i > Y_j \text{ and } \sum_{k=1}^{d} q(i,j,k) \geq 1 \tag{18}$$

$$0 \le v_k \le 1 \qquad\qquad\qquad\qquad \forall k \in \{1, \ldots, d\} \quad (19)$$

$$c_k^{ij} \ge 0 \qquad\qquad \forall i \in \{1, \ldots, n\}, j \in \{1, \ldots, n\}, k \in \{1, \ldots, d\} \quad (20)$$

4:      Identify $i^\star$ such that $v_{i^\star} = \max_i\{v_i\}$, breaking ties arbitrarily. Set $B \leftarrow B \cup \{i_{\max}\}$.
5: **end while**
6: Return $\hat{A} = B$.

---

**Algorithm 3'** is defined to be the batch version of Algorithm 3. Namely, there are $n$ samples in total, divided into $\frac{n}{s}$ batches. The first iteration of the sequential procedure is performed on the first batch, the second iteration on the second batch, and so on.

We are now ready to state the two-stage algorithm for estimating the function $\hat{f}_n$.

---

**Algorithm 4** Two Stage Isotonic Regression (TSIR)

---

**Input:** Values $(X_1, Y_1), \ldots, (X_n, Y_n)$; sparsity level $s$
**Output:** The estimated function, $\hat{f}_n$
1: Estimate $\hat{A}$ by using Algorithm 2, 3, or 3'. Let $v_k = 1$ if $k \in \hat{A}$ and $v_k = 0$ otherwise.
2: Solve the following optimization problem.

$$\min \sum_{i=1}^{n} (Y_i - F_i)^2 \qquad\qquad\qquad\qquad (21)$$

$$\text{s.t.} \sum_{k=1}^{d} q(i,j,k)v_k \ge F_i - F_j \qquad\qquad \forall i, j \in \{1, \ldots, n\} \quad (22)$$

$$F_i \in \mathcal{R} \qquad\qquad \forall i \in \{1, \ldots, n\} \quad (23)$$

     In the Noisy Input Model, replace the objective with $\sum_{i \in \mathcal{S}^+} (1 - F_i) + \sum_{i \in \mathcal{S}^-} F_i$.
3: Return the function $\hat{f}_n(x) = \max\{F_i : X_i \preceq x\}$.

---

Both algorithms for support recovery are linear programs, which can be solved in polynomial time. The second step of Algorithm 4 when applied to the Noisy Output Model is a linearly-constrained quadratic minimization program that can be solved in polynomial time. The following lemma shows that Step 2 of Algorithm 4 when applied to the Noisy Input Model can be reduced to a linear program.

**Lemma 1.** *Under the Noisy Input Model, replacing the constraints $F_i \in \{0, 1\}$ with $F_i \in [0, 1]$ in Problems* (5)-(9) *and* (21)-(23) *does not change the optimal value. Furthermore, there always exists an integer optimal solution that can be constructed from an optimal solution in polynomial time.*

## 3 Results on the Noisy Output Model

Recall the Noisy Output Model: $Y = f(X) + W$, where $f$ is an $s$-sparse coordinate-wise monotone function with active coordinates $A$. We assume throughout this section that $X$ is a uniform random variable on $[0, 1]^d$, $W$ is a zero-mean random variable independent from $X$, and the domain of $f$ is $[0, 1]^d$. We additionally assume that $Y \in [0, 1]$ almost surely. Up to shifting and scaling, this is equivalent to assuming that $f$ has a bounded range and $W$ has a bounded support.

### 3.1 Statistical consistency

In this section, we extend the results of [11], in order to demonstrate the statistical consistency of the estimator produced by Algorithm 1. The consistency will be stated in terms of the $L_2$ norm error.

**Definition 2** ($L_2$ Norm Error). *For an estimator $\hat{f}_n$, define*

$$\|\hat{f}_n - f\|_2^2 \triangleq \int_{x \in [0,1]^d} \left( \hat{f}_n(x) - f(x) \right)^2 dx.$$

*We call $\|\hat{f}_n - f\|_2$ the $L_2$ norm error.*

**Definition 3** (Consistent Estimator). *Let $\hat{f}_n$ be a estimator for the function $f$. We say that $\hat{f}_n$ is consistent if for all $\epsilon > 0$, it holds that*

$$\lim_{n \to \infty} \mathbb{P}\left( \|\hat{f}_n - f\|_2 \ge \epsilon \right) \to 0.$$

**Theorem 1.** *The $L_2$ error of the estimator $\hat{f}_n$ obtained from Algorithm 1 is upper bounded as*

$$\mathbb{P}\left(\|\hat{f}_n - f\|_2 \geq \epsilon\right) \leq 8\binom{d}{s} \exp\left[\left(\frac{128\log(2)}{\epsilon^2} + 2^{\frac{64}{\epsilon^2}} 2^s\right) n^{\frac{s-1}{s}} - \frac{\epsilon^4 n}{512}\right].$$

**Corollary 1.** *When $n = \max\left\{e^{\omega(s^2)}, \omega(s\log(d))\right\}$, the estimator $\hat{f}_n$ from Algorithm 1 is consistent. Namely, $\|\hat{f}_n - f\|_2 \to 0$ in probability as $n \to \infty$. In particular, if the sparsity level $s$ is constant, the sample complexity is only logarithmic in the dimension.*

### 3.2 Support recovery

In this subsection, we give support recovery guarantees for Algorithm 3. The guarantees will be in terms of the values $p_k$, defined below.

**Definition 4.** *Let $Y_1 = f(X_1) + W_1$ and $Y_2 = f(X_2) + W_2$ be two independent samples from the model. For $k \in A$, let*

$$p_k \triangleq \mathbb{P}\left(Y_1 > Y_2 \mid q(1,2,k) = 1\right) - \mathbb{P}\left(Y_1 < Y_2 \mid q(1,2,k) = 1\right).$$

*Assume without loss of generality that $A = \{1, 2, \ldots, s\}$ and $p_1 \leq p_2 \leq \cdots \leq p_s$.*

**Lemma 2.** *It holds that $p_k > 0$ for all $k$. In other words, when $X_1$ is greater than $X_2$ in at least one active coordinate, the output corresponding to $X_1$ is likely to be larger than the one corresponding to $X_2$.*

**Theorem 2.** *Let $B$ be the set of indices corresponding to running Algorithm 3' using $n$ samples. Then it holds that $B = A$ with probability at least*

$$1 - ds\exp\left(-\frac{p_1^2 n}{64 s^3}\right).$$

**Corollary 2.** *Assume that $p_1 = \Theta(1)$. Let $n$ be the number of samples used by Algorithm 3'. If $n = \omega(s^3 \log(d))$, then Algorithm 3' recovers the true support w.h.p. as $n \to \infty$.*

For $x \in \mathbb{R}^d$, let $x_A$ denote $x$ restricted to coordinates defined by the set $A$. Suppose that $s$ is constant, and the sequence of functions $\{f_d\}$ extends a function on $s$ variables, i.e. $f_d$ is defined as $f_d(x) = g(x_A)$ for some $g : [0,1]^s \to \mathcal{R}$. In that case, $p_1 = \Theta(1)$.

We can now give a guarantee of the success of Algorithm 4, using Algorithm 3' for support recovery.

**Corollary 3.** *Assume that $p_1 = \Theta(1)$. Consider running Algorithm 4 using $n$ samples for sequential recovery. Let $m = \frac{n}{s}$. Consider using an additional $m$ samples for function value estimation, so that the total sample size is $n + m$. Let $\hat{f}_{n+m}$ be the estimated function. If $n = \max\left\{\omega(s^3\log(d)), se^{\omega(s^2)}\right\}$, then $\hat{f}_{n+m}$ is a consistent estimator.*

Corollary 3 shows that if $s$ is constant and the sequence of functions $\{f_d\}$ extends a function of $s$ variables, then Algorithm 4 produces a consistent estimator with $n = \omega(\log(d))$ samples. In the supplementary material, we state the statistical consistency results for the Noisy Input Model.

## 4 Experimental results

All algorithms were implemented in Java version 8, using Gurobi version 6.0.0.

### 4.1 Support recovery

We test the support recovery algorithms on random synthetic instances. Let $A = \{1, \ldots, s\}$ without loss of generality. First, randomly sample $r$ "anchor points" in $[0,1]^d$, calling them $Z_1, \ldots, Z_r$. The parameter $r$ governs the complexity of the function produced. In our experiment, we set $r = 10$. Next, randomly sample $X_1, \ldots, X_n$ in $[0,1]^d$. For $i \in \{1, \ldots, n\}$, assign $Y_i = 1 + W_i$ if $Z_j \preceq_A X_i$ for some $j \in \{1, \ldots, r\}$, and assign $Y_i = W_i$ otherwise. The linear programming based algorithms for support recovery, LPSR and S-LPSR, are compared to the simultaneous approach, IPIR, which estimates the active coordinates while also estimating the function values. Note that even though the

proof of support recovery using S-LPSR requires fresh data at each iteration, our experiments do not use fresh data. We keep $s = 3$ fixed and vary $d$ and $n$. The error is Gaussian with mean $0$ and variance $0.1$, independent across coordinates. We report the percentages of successful recovery (see Table 1). The IPIR algorithm performs the best on nearly all settings of $(n, d)$. This suggests that the objective of the IPIR algorithm- to minimize the number of misclassifications on the data- gives the algorithm an advantage in selecting the true active coordinates. The LPSR algorithm outperforms the S-LPSR algorithm when $d = 5$, but the situation reverses for $d \in \{10, 20\}$. For $n = 200$ samples and $d = 5$, the LPSR algorithm correctly recovers the coordinates all but one time, while S-LPSR succeeds $86\%$ of the time. For $d = 10$, LPSR and S-LPSR succeed $46$ and $75\%$ of the time, respectively; for $d = 20$, LPSR and S-LPSR succeed $30$ and $63\%$ of the time, respectively. It appears that determining the coordinates one at a time provides implicit regularization for larger $d$.

We additionally compare the accuracy in function estimation (Table 2). We found these results to be extremely encouraging. For $n = 200$ samples, the IPIR and S-LPSR algorithms had accuracy rates in the range of $87 - 90\%$.

Table 1: Performance of support recovery algorithms on synthetic instances. Each line of the table corresponds to 100 trials.

|       | **IPIR** $d =$ | | | **LPSR** $d =$ | | | **S-LPSR** $d =$ | | |
|-------|----|----|----|----|----|----|----|----|----|
| $n$   | 5  | 10 | 20 | 5  | 10 | 20 | 5  | 10 | 20 |
| 50    | 62 | 55 | 57 | 76 | 29 | 1  | 62 | 33 | 26 |
| 100   | 92 | 85 | 90 | 92 | 33 | 13 | 76 | 56 | 49 |
| 150   | 98 | 94 | 91 | 99 | 50 | 16 | 86 | 71 | 65 |
| 200   | 95 | 99 | 92 | 99 | 46 | 30 | 86 | 75 | 63 |

Table 2: Accuracy of isotonic regression on synthetic instances. Each line of the table corresponds to 100 trials.

|       | **IPIR** $d =$ | | | **LPSR** $d =$ | | | **S-LPSR** $d =$ | | |
|-------|------|------|------|------|------|------|------|------|------|
| $n$   | 5    | 10   | 20   | 5    | 10   | 20   | 5    | 10   | 20   |
| 50    | 78.2 | 77.8 | 77.6 | 77.4 | 74.2 | 65.9 | 77.1 | 76.1 | 74.3 |
| 100   | 85.1 | 85.8 | 84.6 | 84.1 | 77.6 | 75.0 | 84.2 | 83.9 | 81.7 |
| 150   | 87.9 | 87.8 | 86.8 | 87.8 | 81.3 | 77.9 | 87.1 | 86.6 | 85.9 |
| 200   | 89.2 | 89.8 | 88.3 | 89.1 | 83.6 | 83.4 | 89.0 | 88.9 | 87.5 |

## 4.2   Cancer classification using gene expression data

The presence or absence of a disease is believed to follow a monotone relationship with respect to gene expression. Similarly, classifying patients as having one of two diseases amounts to applying the monotonicity principle to a subpopulation of individuals having one of the two diseases. In order to assess the applicability of our sparse monotone regression approach, we apply it to cancer classification using gene expression data. The motivation for using a *sparse* model for disease classification is that certain genes should be more responsible for disease than others. Sparsity can be viewed as a kind of regularization; to prevent overfitting, we allow the regression to explain the results using only a small number of genes.

The data is drawn from the COSMIC database [9], which is widely used in quantitative research in cancer biology. Each patient in the database is identified as having a certain type of cancer. For each patient, gene expressions of tumor cells are reported as a z-score. Namely, if $\mu_G$ and $\sigma_G$ are the mean and standard deviation of the gene expression of gene $G$ and $x$ is the gene expression of a certain patient, then his or her z-score would be equal to $\frac{x - \mu_G}{\sigma_G}$. We filter the patients by cancer type, selecting those with skin and lung cancer, two common cancer types. There are 236698 people with lung or skin cancer in the database, though the database only includes gene expression data for 1492 of these individuals. Of these, 1019 have lung cancer and 473 have skin cancer. A classifier always

selecting "lung" would have an expected correct classification rate of $1019/1492 \approx 68\%$. Therefore this rate should be regarded as the baseline classification rate.

Our goal is to use gene expression data to classify the patients as having either skin or lung cancer. We associate skin cancer as a "0" label and lung cancer as a "1" label. We only include the 20 most associated genes for each of the two types, according to the COSMIC website. This leaves 31 genes, since some genes appear on both lists. We additionally include the negations of the gene expression values as coordinates, since a lower gene expression of certain genes may promote lung cancer over skin cancer. The number of coordinates is therefore equal to 62. The number of active genes is ranged between 1 and 5.

We perform both simultaneous and two-stage isotonic regression, comparing the IPIR and TSIR algorithms, using S-LPSR to recover the coordinates in the two-stage approach. Since for every gene, its negation also corresponds to a coordinate, we added additional constraints. In IPIR, we use variables $v_k \in \{0, 1\}$ to indicate whether coordinate $k$ is in the estimated set of active coordinates. In LPSR and S-LPSR, we use variables $v_k \in [0, 1]$ instead. In order to incorporate the constraints regarding negation of coordinates in IPIR, we included the constraint $v_i + v_j \leq 1$ for pairs $(i, j)$ such that coordinate $j$ is the negation of coordinate $i$. In S-LPSR, once a coordinate $v_i$ was selected, its negation was set to zero in future iterations. The LPSR algorithm, however, could not be modified to take this additional structure into account without using integer variables. Adding the constraints $v_i + v_j \leq 1$ when coordinate $j$ is the negation of coordinate $i$ proved to be insufficient. Therefore, we do not include the LPSR algorithm in our experiments on the COSMIC database.

We compare our isotonic regression algorithms to two classical algorithms: $k$-Nearest Neighbors ([8]) and the Support Vector Machine ([4]). Given a test sample $x$ and an odd number $k$, the $k$-Nearest Neighbors algorithm finds the $k$ closest training samples to $x$. The label of $x$ is chosen according to the majority of the labels of the $k$ closest training samples. The SVM algorithm used is the soft-margin classifier with penalty $C$ and polynomial kernel given by $K(x, y) = (1 + x \cdot y)^m$. We have additionally implemented a version of kNN with dimensionality reduction, in an effort to reduce the curse-of-dimensionality suffered by kNN. Data points are compressed by Principal Component Analysis ([18]) prior to nearest-neighbor classification. However, this version of kNN performed worse than the basic version, and we omit the results.

In Table 3, each row is based on 10 trials, with 1000 test data points chosen uniformly and separately from the training points. The two-stage method was generally faster than the simultaneous method. With 200 training points and $s = 3$, the simultaneous method took 260 seconds on average per trial, while the two-stage method took only 42 seconds per trial. The simultaneous method became prohibitively slow for higher values of $n$. The averages for $k$-Nearest Neighbors and Support Vector Machine are taken as the best over parameter choices in hindsight. For $k$-Nearest Neighbors, $k \in \{1, 3, 5, 7, 9, 11, 15\}$, and for SVM, $C \in \{10, 100, 500, 1000\}$ and $m \in \{1, 2, 3, 4\}$. The fact that the sparse isotonic regression method outperforms the $k$-NN classifier and the polynomial kernel SVM by such a large margin can be explained by a difference in structural assumptions; the results suggest that monotonicity, rather than proximity or a polynomial functional relationship, is the correct property to leverage.

Table 3: Comparison of classifier success rates on COSMIC data. Top row data is according to the "min" interpolation rule and bottom row data is according to the "max" interpolation rule.

| $n$ | IPIR $s =$ | | | | | TSIR + S-LPSR $s =$ | | | | | $k$-NN | SVM |
|---|---|---|---|---|---|---|---|---|---|---|---|---|
| | 1 | 2 | 3 | 4 | 5 | 1 | 2 | 3 | 4 | 5 | | |
| 100 | 83.1 | 84.6 | 76.8 | 66.2 | 53.8 | 82.4 | 84.6 | 77.8 | 73.0 | 65.4 | 69.8 | 63.8 |
| | 83.9 | 91.8 | 91.0 | 85.7 | 75.7 | 82.9 | 90.4 | 88.9 | 87.4 | 83.3 | | |
| 200 | 85.4 | 88.1 | 84.3 | 73.9 | 62.7 | 85.4 | 89.3 | 86.7 | 81.2 | 76.9 | 76.6 | 72.6 |
| | 85.8 | 92.6 | 96.4 | 88.9 | 83.9 | 85.8 | 94.5 | 95.9 | 95.3 | 93.0 | | |
| 300 | - | - | - | - | - | 84.7 | 91.7 | 89.0 | 84.4 | 80.2 | 76.6 | 74.2 |
| | - | - | - | - | - | 85.1 | 94.2 | 95.6 | 95.9 | 94.8 | | |
| 400 | - | - | - | - | - | 85.6 | 91.8 | 89.7 | 87.3 | 81.7 | 78.6 | 77.4 |
| | - | - | - | - | - | 85.8 | 94.0 | 95.7 | 96.4 | 95.7 | | |

The results suggest that the correct sparsity level is $s = 3$. With $n = 400$ samples, the classification accuracy rate is $95.7\%$. When the sparsity level is too low, the monotonicity model is too simple to accurately describe the monotonicity pattern. On the other hand, when the sparsity level is too high, fewer points are comparable, which leads to fewer monotonicity constraints. For $n \in \{100, 200\}$ and $d \in \{1, 2, 3, 4, 5\}$, TSIR + S-LPSR does at least as well as IPIR on 15 out of 20 of $(n, d)$ pairs, and outperforms on 12 of these. This result is surprising, because synthetic experiments show that IPIR outperforms S-LPSR on support recovery.

We further investigate the TSIR + S-LPSR algorithm. Figure 1 shows how the two-stage procedure labels the training points. The high success rate of the sparse isotonic regression method suggests that this nonlinear picture is quite close to reality. The observed clustering of points may be a feature of the distribution of patients, or could be due to a saturation in measurement. Figure 2 studies the robustness of TSIR + S-LPSR. Additional synthetic zero-mean Gaussian noise is added to the inputs, with varying standard deviation. The "max" classification rule is used. 200 training points and 1000 test points were used. Ten trials were run, with one standard deviation error bars indicated in gray. The results indicate that TSIR + S-LPSR is robust to moderate levels of noise.

We note that because the gene expression is measured from tumor cells, much of the variation between the lung and skin cancer patients can be attributed to intrinsic differences between lung and skin cells. Still, this classification task is highly non-linear and challenging, as evidenced by the poor performance of other classifiers. We view these experiments as a proof-of-concept, showing that our algorithm can perform well on real data. An example of a more medically relevant application of our algorithm would be identifying patients as having cancer or not, using blood protein levels [3].

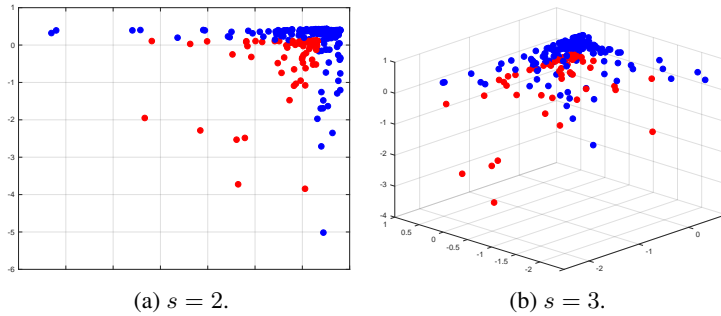

(a) $s = 2$.          (b) $s = 3$.

Figure 1: Illustration of the TSIR + S-LPSR algorithm. Blue and red markers correspond to lung and skin cancer, respectively.

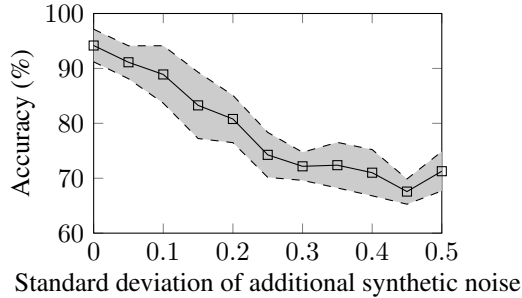

Figure 2: Robustness to error of TSIR + S-LPSR.

## 5  Conclusion

In this paper, we have considered the sparse isotonic regression problem under two noise models: Noisy Output and Noisy Input. We have formulated optimization problems to recover the active coordinates, and then estimate the underlying monotone function. We provide explicit guarantees on the performance of these estimators. We leave the analysis of Linear Programming Support Recovery (Algorithm 2) as an open problem. Finally, we demonstrate the applicability of our approach to a cancer classification task, showing that our methods outperform widely-used classifiers. While the task of classifying patients with two cancer types is relatively simple, the accuracy rates illustrate the modeling power of the sparse monotone regression approach.

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
