[Supplementary Material · supplementary-material.pdf]

*Proof of Lemma 1.* Consider Problem (21)-(23), with the objective function replaced by $\sum_{i \in \mathcal{S}^+} (1 - F_i) + \sum_{i \in \mathcal{S}^-} F_i$. Here, the vector $v$ is fixed. Since $F_i \in [0, 1]$ for all $i \in \{1, \ldots, n\}$, $F_i - F_j \in [-1, 1]$. The left side of Constraint (22) takes value in $\{0, 1, \ldots, s\}$. Therefore, the constraint is tight only when the left side is equal to 0. Therefore, we only require that for $(i, j)$ such that $\sum_{k=1}^d q(i, j, k) v_k = 0$, it holds that $F_i \le F_j$. Let $\epsilon = \min \{\min_{F_i > 0} \{F_i\}, \min_{F_i < 1} \{1 - F_i\}\}$. In other words, $\epsilon$ is the margin to the endpoints of $[0, 1]$. Suppose that $F$ is an optimal solution with some values $F_i \in (0, 1)$. Then $\epsilon > 0$. Let $C = \{i : F_i \in (0, 1)\}$. Consider adding $\epsilon$ to each $F_i$ such that $i \in C$, and call the new solution $F^{+\epsilon}$. Clearly, $F^{+\epsilon}$ is feasible. The change in the objective is equal to

$$\sum_{i \in \mathcal{S}^+} \left(1 - F_i^{+\epsilon}\right) + \sum_{i \in \mathcal{S}^-} F_i^{+\epsilon} - \sum_{i \in \mathcal{S}^+} (1 - F_i) - \sum_{i \in \mathcal{S}^-} F_i$$
$$= \epsilon \left( \left| \{i : i \in C, i \in \mathcal{S}^-\} \right| - \left| \{i : i \in C, i \in \mathcal{S}^+\} \right| \right)$$

On the other hand, consider subtracting $\epsilon$ from each $F_i$ such that $i \in C$, and call the new solution $F^{-\epsilon}$. By construction, $F^{-\epsilon}$ is also feasible. The chance in the objective is equal to $\epsilon \left( \left| \{i : i \in C, i \in \mathcal{S}^+\} \right| - \left| \{i : i \in C, i \in \mathcal{S}^-\} \right| \right)$. Since we have assumed that $F$ is an optimal solution, both changes must be nonnegative. Since they are negations of each other, they must both be equal to zero. Therefore, the solutions $F^{+\epsilon}$ and $F^{-\epsilon}$ have the same objective value as the solution $F$. If $\epsilon = \min_{F_i > 0} \{F_i\}$, choose $F^{-\epsilon}$, and if $\epsilon = \min_{F_i < 1} \{1 - F_i\}$, choose $F^{+\epsilon}$. This leads to the size of the set $C$ decreasing by at least one. Repeating this process inductively, we eventually produce a solution with $C = \emptyset$. Therefore, we have shown that there always exists an integer optimal solution.

We have shown that for fixed $v$, there always exists an integer optimal solution. Furthermore, the process of converting an optimal solution into an integer optimal solution runs in polynomial time. Varying $v$, it also holds that the optimal solutions to Problem (5)-(9) remains the same. The same procedure applies to convert an optimal solution into an integer optimal solution in polynomial time. $\qquad\square$

# A   Results on the Noisy Input Model

Recall the Noisy Input Model: $Y = f(X + W)$, where $f$ is an $s$-sparse coordinate-wise monotone function with active coordinates $A$. We assume throughout this section that $X$ is a uniform random variable on $[0, 1]^d$, $W$ is a zero-mean random variable independent from $X$ with independent coordinates, and $f : \mathbb{R}^d \to \{0, 1\}$.

In this section, we prove the statistical consistency of Two-Stage Isotonic Regression, with Sequential Linear Programming Support Recovery as the support recovery algorithm. In Subsection A.1 we consider the setting where the set of active coordinates is known, and provide an upper bound on the resulting $L_2$-norm error of our estimator. In Subsection A.2 we provide a guarantee on the probability of correctly estimating the support, using S-LPSR. These results are combined to give Corollary 7, stated at the end of the section. As a special case of the corollary, if $s$ is constant and the sequence of functions $\{f_d\}$ extends a function of $s$ variables, and $n = \omega(\log(d))$ samples are used by TSIR, then the estimator $\hat{f}_n$ that is produced is consistent.

## A.1   Statistical consistency

Suppose that the set of active coordinates, $A$, is known. Then we can apply Problem (21)-(23) within Algorithm 4 to estimate the function values, with the variables $v_i$ that indicate the active coordinates set to 1 if $i \in A$, and set to 0 otherwise. The coordinates outside the active set do not influence the solution of the optimization problem, and therefore do not affect the estimated function. Therefore, the setting where $A$ is known is equivalent to the non-sparse setting with dimension $d = s$.

We investigate the regime under which Problem (21)-(23) produces a consistent estimator, in the non-sparse setting ($d = s$). To state our guarantees, it is convenient to represent binary coordinate-wise monotone functions in terms of *monotone partitions*.

**Definition 5** (Monotone Partition). *We say that $(S_0, S_1)$ is a monotone partition of $\mathbb{R}^d$ if*

1. *$S_0$ and $S_1$ form a partition of $\mathbb{R}^d$. That is, $S_0 \cup S_1 = \mathbb{R}^d$ and $S_0 \cap S_1 = \emptyset$.*

2. *For all $x, y \in \mathbb{R}^d$, if $x \preceq y$, then either (i) $x, y \in S_0$, (ii) $x, y \in S_1$, or (iii) $x \in S_0, y \in S_1$.*

*Let $\mathcal{M}_d$ be the set of all monotone partitions of $\mathbb{R}^d$.*

Note that there is a one-to-one correspondence between monotone partitions and binary coordinate-wise monotone functions.

Let $Y = f(X + W)$ represent our model, with $d = s$, and with $f$ corresponding to a monotone partition $(S_0^\star, S_1^\star)$. That is, $f(x) = 0$ for $x \in S_0^\star$ and $f(x) = 1$ for $x \in S_1^\star$. Let $h_0(x)$ be the probability density function of $X$, conditional on $Y = 0$. Similarly, let $h_1(x)$ be the probability density function of $X$, conditional on $Y = 1$. For $(S_0, S_1) \in \mathcal{M}_d$, let

$$H_0(S_1) = \int_{z \in S_1} h_0(z) dz \ \text{ and } \ H_1(S_0) = \int_{z \in S_0} h_1(z) dz.$$

Finally, let $p$ be the probability that $Y = 0$. Let

$$q(S_0, S_1) \triangleq p H_0(S_1) + (1-p) H_1(S_0).$$

The value of $q(S_0, S_1)$ is the probability of misclassification, under the monotone partition $(S_0, S_1)$.

**Assumption 2.** *We assume that $q$ has a unique minimizer on $\mathcal{M}_d$, which is $(S_0^\star, S_1^\star)$.*

**Definition 6** (Discrepancy). *For two monotone partitions $(S_0, S_1)$ and $(S_0', S_1')$, the discrepancy function $D : \mathcal{M}_d \times \mathcal{M}_d \to [0, 1]$ is defined as follows.*

$$D\left((S_0, S_1), (S_0', S_1')\right) \triangleq \mathbb{P}\left(X \in S_0 \cap S_1'\right) + \mathbb{P}\left(X \in S_0' \cap S_1\right)$$

*Also let*

$$B_\delta\left(S_0^\star, S_1^\star\right) \triangleq \left\{(S_0, S_1) \in \mathcal{M}_d : D\left((S_0, S_1), (S_0^\star, S_1^\star)\right) \le \delta\right\}$$

*be the set of monotone partitions with discrepancy at most $\delta$ from $(S_0^\star, S_1^\star)$.*

**Theorem 3.** *Let $d = s$. Suppose Assumption 2 holds, and the components of $W$ are independent. Let $\hat{f}_n$ be the estimator derived from Algorithm 1, and let*

$$q_{min}(\delta) \triangleq \min\left\{q(S_0, S_1) : (S_0, S_1) \notin B_\delta(S_0^\star, S_1^\star)\right\} > q(S_0^\star, S_1^\star).$$

*Then for any $0 < \delta \le 1$,*

$$\mathbb{P}\left(\|\hat{f}_n - f\|_2 > \delta\right) \le$$

$$\frac{\exp\left[(2^s + 2\log(2) - 1) n^{\frac{s-1}{s}}\right]}{\exp\left[n^{\frac{2s-1}{2s}}\right]} + \left(\exp\left[n^{\frac{2s-1}{2s}}\right] + 1\right) \exp\left(-\frac{\left(q_{min}(\delta) - q\left(S_0^\star, S_1^\star\right)\right)^2 n}{36}\right).$$

**Corollary 4.** *Suppose that $q_{min}(\delta) - q\left(S_0^\star, S_1^\star\right) = \Theta(1)$, that is, constant in $s$. When $d = s$ and $n = e^{\omega(s^2)}$, the estimator $\hat{f}_n$ produced by Algorithm 1 is consistent.*

Theorem 3 has an analogous version in the sparse setting ($s < d$). First we need some definitions, similar to those that precede Theorem 3. We write $x =_A y$ if $x \preceq_A y$ and $x \succeq_A y$.

**Definition 7** (s-Sparse Monotone Partition). *We say that $(S_0, S_1)$ is an $s$-sparse monotone partition of $\mathbb{R}^d$ if*

1. *$S_0$ and $S_1$ form a partition of $\mathbb{R}^d$. That is, $S_0 \cup S_1 = \mathbb{R}^d$ and $S_0 \cap S_1 = \emptyset$.*

2. *There exists a set $A \subset [d]$ such that for all $x, y \in \mathbb{R}^d$, if $x \preceq_A y$, then either (i) $x, y \in S_0$, (ii) $x, y \in S_1$, or (iii) $x \in S_0, y \in S_1$. Note that this implies that if $x =_A y$, then either $x, y \in S_0$ or $x, y \in S_1$.*

*Let $\mathcal{M}_{s,d}$ be the set of all $s$-sparse monotone partitions of $\mathbb{R}^d$.*

Note that there is a one-to-one correspondence between monotone partitions and $s$-sparse binary coordinate-wise monotone functions.

Let $Y = f(X + W)$ represent our model, with $d < s$, and with $f$ corresponding to an $s$-sparse monotone partition $(S_0^\star, S_1^\star)$. That is, $f(x) = 0$ for $x \in S_0^\star$ and $f(x) = 1$ for $x \in S_1^\star$. Let $h_0(x)$ be

the probability density function of $X$, conditional on $Y = 0$. Similarly, let $h_1(x)$ be the probability density function of $X$, conditional on $Y = 1$. For $(S_0, S_1) \in \mathcal{M}_{s,d}$, let

$$H_0(S_1) = \int_{z \in S_1} h_0(z)dz \text{ and } H_1(S_0) = \int_{z \in S_0} h_1(z)dz.$$

Finally, let $p$ be the probability that $Y = 0$. Let

$$q(S_0, S_1) \triangleq pH_0(S_1) + (1 - p)H_1(S_0).$$

The value of $q(S_0, S_1)$ is the probability of misclassification, under the $s$-sparse monotone partition $(S_0, S_1)$.

**Assumption 3.** *We assume that $q$ has a unique minimizer on $\mathcal{M}_{s,d}$, which is $(S_0^\star, S_1^\star)$.*

**Definition 8** (Discrepancy). *For two $s$-sparse monotone partitions $(S_0, S_1)$ and $(S_0', S_1')$, the discrepancy function $D : \mathcal{M}_{s,d} \times \mathcal{M}_{s,d} \to [0, 1]$ is defined as follows.*

$$D((S_0, S_1), (S_0', S_1')) \triangleq \mathbb{P}(X \in S_0 \cap S_1') + \mathbb{P}(X \in S_0' \cap S_1)$$

*Also let*

$$B_\delta^s(S_0^\star, S_1^\star) \triangleq \{(S_0, S_1) \in \mathcal{M}_{s,d} : D((S_0, S_1), (S_0^\star, S_1^\star)) \leq \delta\}$$

*be the set of $s$-sparse monotone partitions with discrepancy at most $\delta$ from $(S_0^\star, S_1^\star)$.*

**Theorem 4.** *Suppose Assumption 3 holds, and the components of $W$ are independent. Let $\hat{f}_n$ be the estimator derived from Algorithm 1 and let*

$$q_{min}(\delta) \triangleq \min\{q(S_0, S_1) : (S_0, S_1) \notin B_\delta^s(S_0^\star, S_1^\star)\} > q(S_0^\star, S_1^\star)\}.$$

*Then for any $0 < \delta \leq 1$,*

$$\mathbb{P}\left(\|\hat{f}_n - f\|_2 > \delta\right) \leq$$

$$\frac{\exp\left[(2^s + 2\log(2) - 1)n^{\frac{s-1}{s}}\right]}{\exp\left[n^{\frac{2s-1}{2s}}\right]} + \left(\binom{d}{s}\exp\left[n^{\frac{2s-1}{2s}}\right] + 1\right)\exp\left(-\frac{(q_{min}(\delta) - q(S_0^\star, S_1^\star))^2 n}{36}\right).$$

Theorem 3 allows us to state the following corollary regarding the IPIR algorithm.

**Corollary 5.** *Suppose $s$ is constant and the sequence of functions $\{f_d\}$ extends a function of $s$ variables. Let $\hat{f}_n$ be the estimator produced by Algorithm 1. If $n = \omega(\log(d))$, then $\hat{f}_n$ is a consistent estimator.*

## A.2 Support recovery

In this subsection, we give support recovery guarantees for Algorithm 3'. The guarantees will be in terms of differences of probabilities.

**Definition 9.** *Let $Y_1 = f(X_1 + W_1)$ and $Y_2 = f(X_2 + W_2)$ be two independent samples from the model. For $k \in A$, define*

$$\bar{p}_k \triangleq \mathbb{P}(Y_1 = 1, Y_2 = 0 \mid q(1, 2, k) = 1) - \mathbb{P}(Y_1 = 0, Y_2 = 1 \mid q(1, 2, k) = 1).$$

*Assume without loss of generality that $A = \{1, \ldots, s\}$ and $\bar{p}_1 \leq \bar{p}_2 \leq \cdots \leq \bar{p}_s$.*

**Lemma 3.** *For all $k \in A$, it holds that $\bar{p}_k > 0$.*

**Theorem 5.** *Let $B$ be the set of indices corresponding to running Algorithm 3' using $n$ samples. Then it holds that $B = A$ with probability at least*

$$1 - ds \exp\left(-\frac{\bar{p}_1^2 n}{64s^3}\right).$$

We can now give a guarantee of the success of Algorithm 4, using Algorithm 3 for support recovery.

**Corollary 6.** *Assume that $\bar{p}_1 = \Theta(1)$. Let $n$ be the number of samples used by Algorithm 3'. If $n = \omega(s^3 \log(d))$, then Algorithm 3' recovers the true support w.h.p. as $n \to \infty$.*

**Corollary 7.** *Suppose that $q_{min}(\delta) - q(S_0^\star, S_1^\star) = \Theta(1)$. Suppose also that $\bar{p}_1 = \Theta(1)$, and that the components of $W$ are independent. Consider running Algorithm 4 using $n$ samples for sequential support recovery. Let $m = \frac{n}{s}$. Consider using an additional $m$ samples for function value estimation, so that the total number of samples is $n+m$. Let $\hat{f}_{n+m}$ be the estimated function. If $n = \omega(s^3 \log(d))$ and $n = se^{\omega(s^2)}$, then $\hat{f}_{n+m}$ is a consistent estimator.*

# B Proofs for the Noisy Output Model

We will build toward a proof of Theorem 1. We note that Algorithm 1 selects an $s$-sparse coordinate-wise monotone function that minimizes the empirical $L_2$ loss. To prove the statistical consistency of the estimated function, we need to introduce the expected VC dimension ([24]). Let $\mathcal{F}_{s,d}$ be the set of $s$-sparse coordinate-wise monotone functions on $[0,1]^d$. Following [11], let $Q(x,y,f) = (y - f(x))^2$ for $x \in [0,1]^d$, $y \in \mathbb{R}$, and $f \in \mathcal{F}_{s,d}$. For a fixed sequence $(x_1, y_1), \ldots, (x_n, y_n) \in [0,1]^d \times [0,1]$, consider the set of vectors $\mathbf{Q} = \{(Q(x_1, y_1, f), \ldots, Q(x_n, y_n, f)), f \in \mathcal{F}_{s,d}\}$. In other words, we vary over $\mathcal{F}_{s,d}$ and produce the associated error vectors. Let $N(\epsilon, \mathcal{F}_{s,d}, (x_1, y_1), \ldots, (x_n, y_n))$ be the size of the minimal $\epsilon$-net of the set $\mathbf{Q}$. Namely, a set $E$ is an $\epsilon$-net for $\mathbf{Q}$ if for every $q \in \mathbf{Q}$ there exists $v \in E$ such that $\|q - v\|_\infty \leq \epsilon$. For any $\epsilon > 0$, the *expected VC entropy* of $\mathcal{F}_s$ is defined as

$$N_{\mathcal{F}_{s,d}}(\epsilon, n) = \mathbb{E}\left[N\left(\epsilon, \mathcal{F}_{s,d}, (X_1, Y_1), \ldots, (X_n, Y_n)\right)\right].$$

The expectation is over the random variables $(X_i, Y_i)$. The expected VC entropy measures the complexity of the class $\mathcal{F}_{s,d}$, and can be used to prove convergence in $L_2$.

The following proposition follows from Corollary 1 (pp. 45) of [14].

**Proposition 1.**

$$\mathbb{P}\left(\sup_{\hat{f} \in \mathcal{F}_{s,d}} \left|\int Q(x,y,\hat{f})dF(x,y) - \frac{1}{n}\sum_{i=1}^{n} Q(X_i, Y_i, \hat{f})\right| > \epsilon\right) \leq 4N_{\mathcal{F}_{s,d}}\left(\frac{\epsilon}{16}, n\right) \exp\left(-\frac{\epsilon^2 n}{128}\right).$$

**Proposition 2.** *If* $Y = f(X) + W \in [0,1]$ *almost surely, then*

$$\mathbb{P}\left(\|\hat{f}_n - f\|_2 > \epsilon\right) \leq 8N_{\mathcal{F}_{s,d}}\left(\frac{\epsilon^2}{32}, n\right) \exp\left(-\frac{\epsilon^4 n}{512}\right).$$

*Proof.* Equivalently, we show

$$\mathbb{P}\left(\|\hat{f}_n - f\|_2^2 > \epsilon\right) \leq 8N_{\mathcal{F}_{s,d}}\left(\frac{\epsilon}{32}, n\right) \exp\left(-\frac{\epsilon^2 n}{512}\right).$$

As shown by [12],

$$\|\hat{f}_n - f\|_2^2 = \int Q(x, y, \hat{f}_n)dF(x,y) - \int Q(x,y,f)dF(x,y).$$

Therefore,

$$\mathbb{P}\left(\|\hat{f}_n - f\|_2^2 > \epsilon\right) = \mathbb{P}\left(\int Q(x, y, \hat{f}_n)dF(x,y) - \int Q(x,y,f)dF(x,y) > \epsilon\right).$$

By optimality of $\hat{f}_n$, it holds that $\sum_{i=1}^{n} Q(X_i, Y_i, f) - \sum_{i=1}^{n} Q(X_i, Y_i, \hat{f}_n) \geq 0$. We therefore have

$$\mathbb{P}\left(\|\hat{f}_n - f\|_2^2 > \epsilon\right)$$
$$\leq \mathbb{P}\left(\int Q(x, y, \hat{f}_n)dF(x,y) - \sum_{i=1}^{n} Q(X_i, Y_i, \hat{f}_n) + \sum_{i=1}^{n} Q(X_i, Y_i, f) - \int Q(x,y,f)dF(x,y) > \epsilon\right).$$

Grouping the first two terms and the last two terms, we obtain by the Union Bound,

$$\mathbb{P}\left(\|\hat{f}_n - f\|_2^2 > \epsilon\right) \leq \mathbb{P}\left(\int Q(x, y, \hat{f}_n)dF(x,y) - \sum_{i=1}^{n} Q(X_i, Y_i, \hat{f}_n) > \frac{\epsilon}{2}\right)$$
$$+ \mathbb{P}\left(\sum_{i=1}^{n} Q(X_i, Y_i, f) - \int Q(x,y,f)dF(x,y) > \frac{\epsilon}{2}\right)$$
$$\leq \mathbb{P}\left(\left|\int Q(x, y, \hat{f}_n)dF(x,y) - \sum_{i=1}^{n} Q(X_i, Y_i, \hat{f}_n)\right| > \frac{\epsilon}{2}\right)$$

$$+ \mathbb{P}\left( \left| \sum_{i=1}^{n} Q(X_i, Y_i, f) - \int Q(x, y, f) dF(x, y) \right| > \frac{\epsilon}{2} \right)$$

$$\leq 2\mathbb{P}\left( \sup_{\hat{f} \in \mathcal{F}_{s,d}} \left| \int Q(x, y, \hat{f}) dF(x, y) - \sum_{i=1}^{n} Q(X_i, Y_i, \hat{f}) \right| > \frac{\epsilon}{2} \right)$$

$$\leq 8 N_{\mathcal{F}_{s,d}}\left( \frac{\epsilon}{32}, n \right) \exp\left( -\frac{\epsilon^2 n}{512} \right),$$

where the last inequality follows from Proposition 1. $\qquad\square$

Therefore, if the expected VC entropy of $\mathcal{F}_{s,d}$ grows subexponentially in $n$, the estimator $\hat{f}_n$ derived from Algorithm 1 converges to the true function in $L_2$. Define the non-sparse class $\mathcal{F}_d = \mathcal{F}_{d,d}$.

**Proposition 3.** $N_{\mathcal{F}_{s,d}}(\epsilon, n) \leq \binom{d}{s} N_{\mathcal{F}_s}(\epsilon, n)$.

*Proof.* The set $\mathcal{F}_{s,d}$ can be written as a union of $\binom{d}{s}$ function classes, depending on which subset of the coordinates is active. $\qquad\square$

Our goal is now to bound the expected VC entropy of the class $\mathcal{F}_d$. The expected VC entropy is related to a combinatorial quantity known as the *labeling number*.

**Definition 10** (Labeling Number ([11])). *For a sequence of points $x_1, \ldots, x_n \in [0,1]^d$ and a positive integer $m$, the labeling number $L(m, x_1, \ldots, x_n)$ is the number of functions $\phi : \{x_1, \ldots, x_n\} \rightarrow \{1, 2, \ldots, m\}$ such that $\phi(X_i) \leq \phi(X_j)$ whenever $x_i \preceq x_j$, for $i, j \in \{1, \ldots, n\}$.*

**Proposition 4.** *For any $(x_1, y_1), \ldots, (x_n, y_n) \in [0,1]^d \times [0,1]$,*

$$N\left( \epsilon, \mathcal{F}_d, (x_1, y_1), \ldots, (x_n, y_n) \right) \leq L\left( \left\lceil \frac{2}{\epsilon} \right\rceil, x_1, \ldots, x_n \right).$$

*Let $\overline{\mathcal{F}}_d$ be the set of coordinate-wise monotone functions $f : [0,1]^d \rightarrow \{0,1\}$. Then*

$$N\left( \epsilon, \overline{\mathcal{F}}_d, (x_1, y_1), \ldots, (x_n, y_n) \right) \geq L\left( \left\lfloor \sqrt{\frac{3}{2\epsilon}} \right\rfloor - 3, x_1, \ldots, x_n \right).$$

*Proof.* For the lower bound, let $\delta = \sqrt{\frac{2\epsilon}{3}}$, and let $N = \lfloor \frac{1}{\delta} \rfloor - 3$. Define the sequence $q_i = \delta(i+1)$, for $i \in \{1, \ldots, N\}$. The monotone labelings supported on $\{q_1, \ldots, q_N\}$ are a subset of the coordinate-wise monotone functions. Our goal is to show that for every two distinct labelings $l_1$ and $l_2$, it holds that

$$\| (Q(x_1, y_1, l_1), \ldots, Q(x_n, y_n, l_1)), (Q(x_1, y_1, l_2), \ldots, Q(x_n, y_n, l_2)) \|_\infty > 2\epsilon.$$

If this relation holds for all distinct pairs of labelings, then at least $L(N, x_1, x_2, \ldots, x_n)$ points are required to form an $\epsilon$-net of the set **Q**.

If $l_1$ and $l_2$ are distinct labelings, then there exists $k \in \{1, \ldots, n\}$ such that $l_1(x_k) \neq l_2(x_k)$. Therefore,

$$
\begin{aligned}
|Q(x_k, y_k, l_1) - Q(x_k, y_k, l_2)| &= \left| (l_1(x_k) - y_k)^2 - (l_2(x_k) - y_k)^2 \right| \\
&= \left| l_1(x_k)^2 - 2y_k l_1(x_k) + 2y_k l_2(x_k) - l_2(x_k)^2 \right| \\
&= \left| 2y_k (l_1(x_k) - l_2(x_k)) + (l_1(x_k) - l_2(x_k))(l_1(x_k) + l_2(x_k)) \right| \\
&= |l_1(x_k) - l_2(x_k)| \, |l_1(x_k) + l_2(x_k) - 2y_k| \\
&\geq \delta \cdot 2 \left| \frac{l_1(x_k) + l_2(x_k)}{2} - y_k \right| \\
&\geq \delta \cdot 2 \min\{q_1, 1 - q_N\} \\
&\geq 4\delta^2 \\
&= \frac{8}{3} \epsilon
\end{aligned}
$$

$$> 2\epsilon$$

We conclude that $L(N, \overline{\mathcal{F}}_d, x_1, \ldots, x_n) \le N\left(\epsilon, \overline{\mathcal{F}}_d, (x_1, y_1), \ldots, (x_n, y_n)\right)$.

For the upper bound, the proof comes from the proof of Proposition 3 in [11]. Let $N = \left\lceil \frac{2}{\epsilon} \right\rceil$. Let $q_i = \frac{i-1}{N}$ for $i \in \{1, \ldots, N, N+1\}$. Define

$$G \triangleq \{((y_1 - g_1)^2, (y_2 - g_2)^2, \ldots, (y_n - g_n)^2) : g_i \in \{q_1, \ldots, q_N\}, x_i \preceq x_j \implies g_i \le g_j\}.$$

Then $|G| \le L(N, x_1, \ldots, x_n)$. We now show that $G$ is an $\epsilon$-net of $\mathbf{Q} = \{(Q(x_1, y_1, f), \ldots, Q(x_n, y_n, f)), f \in \mathcal{F}\}$. For each sample $i \in \{1, \ldots, n\}$, find $k_i$ such that $f(x_i) \in [q_{k_i}, q_{k_i+1})$. Set $g_i = q_{k_i}$. Now,

$$\begin{aligned}
\left|(y_i - f(x_i))^2 - (y_i - q_{k_i})^2\right| &= \left|y_i^2 - 2y_i f(x_i) + f(x_i)^2 - y_1^2 + 2y_i q_{k_i} - q_{k_i}^2\right| \\
&= \left|f(x_i)^2 + 2y_i(q_{k_i} - f(x_i)) - q_{k_i}^2\right| \\
&= |(f(x_i) - q_{k_i})(f(x_i) + q_{k_i}) - 2y_i(f(x_i) - q_{k_i})| \\
&= (f(x_i) - q_{k_i})|f(x_i) + q_{k_i} - 2y_i| \\
&\le \frac{2}{N} \\
&= \frac{2}{\left\lceil \frac{2}{\epsilon} \right\rceil} \\
&\le \epsilon
\end{aligned}$$

It remains to show that $x_i \preceq x_j \implies g_i \le g_j$. Since $f$ is coordinate-wise monotone, $x_i \preceq x_j \implies f(x_i) \le f(x_j)$. Then also $g_i \le g_j$. Therefore, we have shown that $G$ is a valid $\epsilon$-net, and we conclude that the size of the smallest $\epsilon$-net is at most $L(N, x_1, \ldots, x_n)$. $\qquad\square$

The $m$-labeling number is in turn related to the binary labeling number.

**Proposition 5.** *[11] It holds that* $L(m, x_1, \ldots, x_n) \le (L(2, x_1, \ldots, x_n))^{m-1}$.

*Proof.* The proof can be found in the proof of Lemma 3 in [11], with the correction that

$$g_2(x_i) = \begin{cases} 1 & \text{if } g(x_i) \le m \\ 2 & \text{if } g(x_i) = m + 1. \end{cases}$$

$\qquad\square$

Propositions 4 and 5 suggest that the binary labeling number is a good proxy for the VC entropy.

**Theorem 6.** *Let* $X_1, \ldots, X_n$ *be distributed uniformly and independently in* $[0, 1]^d$. *Let* $L(X_1, \ldots, X_n)$ *be the number of binary monotone labelings of the points* $X_1, \ldots, X_n$. *Then for* $k \ge 1$,

$$\exp\left[\frac{\log(2)(1 - e^{-1})k}{(d-1)!}n^{\frac{d-1}{d}}\right] \le \mathbb{E}[L(X_1, \ldots X_n)^k] \le \exp\left[\left(2\log(2)k + 2^{k+d}\right)n^{\frac{d-1}{d}}\right].$$

*We also have*

$$\mathbb{E}[L(X_1, \ldots X_n)^k] \le \exp\left[\left(2^d + 2\log(2) - 1\right)n^{\frac{d-1}{d}}\right].$$

In order to prove the upper bound in Theorem 6, we relate the binary labeling number to the number of *integer partitions*.

**Definition 11** (Integer Partition). *An integer partition of dimension* $(d-1)$ *with values in* $\{0, 1, \ldots m\}$, *is a collection of values* $A_{i_1, i_2, \ldots, i_{d-1}} \in \{0, 1, \ldots, m\}$ *where* $i_k \in \{1, \ldots m\}$ *and* $A_{i_1, i_2, \ldots, i_{d-1}} \le A_{j_1, j_2, \ldots, j_{d-1}}$ *whenever* $i_k \le j_k$ *for all* $k \in \{1, \ldots, d-1\}$. *The set of integer partitions of dimension* $(d-1)$ *with values in* $\{0, 1, \ldots m\}$ *is denoted by* $P([m]^d)$.

Figure 3: Illustration of a partition in $d = 2$ with $m = 10$. The partition cells are indicated in gray, and the border cells are marked.

Note: the definition is in terms of $(d-1)$ because when the monotone regression problem is in dimension $d$, we will consider partitions of dimension $(d-1)$. To illustrate the definition, consider setting $d = 2$ (see Figure 3). An integer partition of dimension 1 is an assignment of values $(A_1, A_2, \ldots, A_m)$ that is non-increasing, and each $A_k$ takes value in $\{0, 1, \ldots, m\}$. A 1-dimensional partition can be seen to divide the $m \times m$ grid in a monotonic way. Next we define the concept of a *border cell*.

**Definition 12** (Border Cell). *Label the cells in the $[m]^d$ grid according to cell coordinates, namely entries $(x_1, x_2, \ldots x_d)$, where $x_k \in \{1, \ldots, m\}$ for each $k \in \{1, \ldots, d\}$. For a partition $p \in P([m]^d)$ with entries in $\{1, \ldots, m\}$, consider its values $A_{i_1, i_2, \ldots, i_{d-1}}$. The cells corresponding to the partition (which we call the partition cells) are given by $(x_1, x_2, \ldots, x_{d-1}, x)$, for $x \leq A_{x_1, x_2, \ldots, x_{d-1}}$ and where each $x_k$ ranges in $\{1, \ldots, m\}$. We say that two cells are adjacent if they share a face or a corner. The border cells are defined to be the partition cells that are adjacent to at least one cell that is not a partition cell.*

**Lemma 4.** *The number of border cells in any $(d-1)$-dimensional integer partition with entries from $\{1, \ldots, m\}$ is at most $m^d - (m-1)^d$.*

*Proof.* When $d = 2$, the number of cells on the border of any (1-dimensional) partition with values in $\{1, \ldots, m\}$ is at most $2m - 1 = m^2 - (m-1)^2$, corresponding to a path from $(1, m)$ to $(m, 1)$. When $d = 3$, the number of border cells in any (2-dimensional) partition with values in $\{1, \ldots, m\}$ is at most corresponding to border cells that include $(1, m, m)$ and $(m, 1, 1)$. All partitions with such border cells have the same number of border cells. The simplest of these is the one where each cell is on the perimeter of the cube. The number of border cells in such a partition is equal to $3m^2 - 3m + 1 = m^3 - (m-1)^3$. For general $d$, the number of border cells in a $(d-1)$-dimensional partition taking values in $\{1, \ldots, m\}$ is upper bounded by the total number of cells minus the number of cells in an $(m-1)^d$ grid, in other words, $m^d - (m-1)^d$. $\square$

The key idea of the proof of the upper bound in Theorem 6 comes from the following lemma.

**Lemma 5.** *Let $k \geq 1$ and let $N \sim \text{Binom}\left(n, \frac{m^d - (m-1)^d}{m^d}\right)$. It holds that*

$$\mathbb{E}[L(X_1, \ldots, X_n)^k] \leq \left|P\left([m]^d\right)\right|^k \mathbb{E}[2^{kN}].$$

*Proof.* The idea of the proof comes from the proof of Theorem 13.13 in [6], who showed a similar result for $d = 2$ and $k = 1$. Consider a binary coordinate-wise monotone function $f$, with domain $[0, 1]^d$. Let $S_0 = \{x \in [0, 1]^d : f(x) = 0\}$ and $S_1 = \{x \in [0, 1]^d : f(x) = 1\}$. The number of binary labelings of a set of points $X_1, \ldots, X_n$ is equal to the number of partitions $(S_0, S_1)$ producing distinct labelings. To upper-bound the number of dividing surfaces, we divide the $d$-dimensional cube into an $m^d$ grid, $[m]^d$. That is, each cell in the grid has side length $\frac{1}{m}$. Let $B$ be the intersection of the boundaries of the $S_0$ and $S_1$. For example, if

$$f(x) = \begin{cases} 0 & \text{if } x_1 + x_2 < 1 \\ 1 & \text{if } x_1 + x_2 \geq 1 \end{cases}$$

then $B = \{x : x_1 + x_2 = 1\}$. Now consider the subset of cells that contain at least one element of $B$. These cells are necessarily the border cells of some $(d-1)$-dimensional integer partition with

values from $\{1, \ldots, m\}$. Therefore, we can upper bound the number of labelings as follows. For a boundary $B$ corresponding to a partition $(S_0, S_1)$, let $\overline{B}$ be the border cells of the corresponding integer partition. Define the set $\mathcal{B}$ to contain all such $\overline{B}$, noting that $|\mathcal{B}| = |P([m]^d)|$. Let $N_{\overline{B}}$ be the number of points falling into the cells comprising $\overline{B}$. For each $\overline{B}$ that corresponds to a partition $(S_0, S_1)$, we add a contribution of $2^{N_{\overline{B}}}$. This contribution corresponds to all (valid or invalid) labelings of the points within the border cells. Points outside the border cells are labeled $0$ if they fall in $S_0$ and $1$ if they fall in $S_1$. Since we have potentially overcounted the number of binary labelings by including invalid labelings, we have the following upper bound.

$$L(X_1, \ldots, X_n) \leq \sum_{\overline{B} \in \mathcal{B}} 2^{N_{\overline{B}}}.$$

Therefore, we also have

$$L(X_1, \ldots, X_n)^k \leq \left( \sum_{\overline{B} \in \mathcal{B}} 2^{N_{\overline{B}}} \right)^k.$$

By Jensen's inequality, we have that for $a_i \geq 0$ and $k \geq 1$,

$$\left( \frac{1}{n} \sum_{i=1}^{n} a_i \right)^k \leq \frac{1}{n} \sum_{i=1}^{n} a_i^k.$$

Therefore,

$$\left( \sum_{\overline{B} \in \mathcal{B}} 2^{N_{\overline{B}}} \right)^k \leq |\mathcal{B}|^{k-1} \sum_{\overline{B} \in \mathcal{B}} 2^{k N_{\overline{B}}}$$

and we have

$$L(X_1, \ldots, X_n)^k \leq |\mathcal{B}|^{k-1} \sum_{\overline{B} \in \mathcal{B}} 2^{k N_{\overline{B}}}$$

$$\mathbb{E}\left[ L(X_1, \ldots, X_n)^k \right] \leq \mathbb{E}\left[ |\mathcal{B}|^{k-1} \sum_{\overline{B} \in \mathcal{B}} 2^{k N_{\overline{B}}} \right]$$

$$= |\mathcal{B}|^{k-1} \sum_{\overline{B} \in \mathcal{B}} \mathbb{E}\left[ 2^{k N_{\overline{B}}} \right]$$

From Lemma 4, the number of points in the border cells of a partition with the maximal number of border cells is distributed as a binomial random variable $N$ with parameters $\left( n, \frac{m^d - (m-1)^d}{m^d} \right)$. We therefore have

$$\mathbb{E}\left[ L(X_1, \ldots, X_n)^k \right] \leq |\mathcal{B}|^{k-1} \sum_{\overline{B} \in \mathcal{B}} \mathbb{E}\left[ 2^{k N} \right]$$

$$= |\mathcal{B}|^k \mathbb{E}\left[ 2^{k N} \right]$$

$$= \left| P([m]^d) \right|^k \mathbb{E}\left[ 2^{k N} \right].$$

$\square$

*Proof of Theorem 6.* **Upper bound**
From Lemma 5, we know that

$$\mathbb{E}[L(X_1, \ldots, X_n)^k] \leq \left| P([m]^d) \right|^k \cdot \mathbb{E}[2^{k N}].$$

Now,

$$\mathbb{E}[2^{k N}] = \mathbb{E}[e^{\log(2) k N}] = M_N(\log(2) k),$$

where $M_N(\cdot)$ is the moment-generating function of the random variable $N$. A binomial random variable $Z$ with parameters $(n, p)$ has moment-generating function $M_Z(\theta) = (1 - p + pe^\theta)^n$. Additionally, [17] showed that

$$\left| P\left([m]^d\right) \right| \leq \binom{2m}{m}^{m^{d-2}}.$$

Substituting,

$$\mathbb{E}[L(X_1, \ldots, X_n)^k] \leq \binom{2m}{m}^{km^{d-2}} \left( 1 - \frac{m^d - (m-1)^d}{m^d} + \frac{m^d - (m-1)^d}{m^d} e^{\log(2)k} \right)^n$$

$$\leq \binom{2m}{m}^{km^{d-2}} \left( 1 + 2^k \frac{m^d - (m-1)^d}{m^d} \right)^n$$

$$\leq \left( 2^{2m} \right)^{km^{d-2}} \left( 1 + 2^k \frac{m^d - (m-1)^d}{m^d} \right)^n$$

$$= 2^{2km^{d-1}} \left( 1 + 2^k \frac{m^d - (m-1)^d}{m^d} \right)^n$$

$$= \exp\left[ 2\log(2)km^{d-1} + n\log\left( 1 + 2^k \frac{m^d - (m-1)^d}{m^d} \right) \right]$$

Choosing $m = n^{\frac{1}{d}}$,

$$\mathbb{E}[L(X_1, \ldots, X_n)^k] \leq \exp\left[ 2\log(2)kn^{\frac{d-1}{d}} + n\log\left( 1 + 2^k \frac{n - \left( n^{\frac{1}{d}} - 1 \right)^d}{n} \right) \right]$$

Since $\log(1 + x) \leq x$,

$$\mathbb{E}[L(X_1, \ldots, X_n)^k] \leq \exp\left[ 2\log(2)kn^{\frac{d-1}{d}} + 2^k \left( n - \left( n^{\frac{1}{d}} - 1 \right)^d \right) \right]$$

Applying the Binomial Theorem,

$$n - \left( n^{\frac{1}{d}} - 1 \right)^d = n - \sum_{k=0}^{d} \binom{d}{k} n^{\frac{d-k}{d}} (-1)^k$$

$$= -\sum_{k=1}^{d} \binom{d}{k} n^{\frac{d-k}{d}} (-1)^k$$

$$\leq \sum_{k=1}^{d} \binom{d}{k} \cdot \max_{k \in \{1, \ldots, d\}} n^{\frac{d-k}{d}} (-1)^{k+1}$$

$$= \left( 2^d - 1 \right) n^{\frac{d-1}{d}}$$

Substituting, we obtain

$$\mathbb{E}[L(X_1, \ldots, X_n)^k] \leq \exp\left[ 2\log(2)kn^{\frac{d-1}{d}} + 2^k \left( 2^d - 1 \right) n^{\frac{d-1}{d}} \right]$$

$$\leq \exp\left[ \left( 2\log(2)k + 2^{k+d} \right) n^{\frac{d-1}{d}} \right].$$

The proof for $k = 1$ is similar.

**Lower Bound**

Let $N$ be an integer, which will be specified later. Divide $[0, 1]^d$ into $N^d$ cells of side length $\frac{1}{N}$. The cells are labeled in the natural coordinate system, writing $C = (x_1, \ldots, x_d) \in [N]^d$. We say that two cells are *incomparable* if for all $x \in C_1$ and $y \in C_2$, neither $x \preceq y$ nor $x \succeq y$. Let us find the number of incomparable cells.

**Lemma 6.** *The number of incomparable cells is at least* $\binom{N+d-2}{d-1}$.

*Proof.* Consider any two cells $C_1 = (x_1, x_2, \ldots, x_d)$ and $C_2 = (y_1, y_2, \ldots, y_d)$. If $\sum_{i=1}^{d} x_i = \sum_{i=1}^{d} y_i$, then either $(x_1, \ldots, x_d) = (y_1, \ldots, y_d)$ or $(x_1, \ldots, x_d) \not\preceq (y_1, \ldots, y_d)$ and $(x_1, \ldots, x_d) \not\succeq (y_1, \ldots, y_d)$. Observe that if $(x_1, \ldots, x_d) \not\preceq (y_1, \ldots, y_d)$ and $(x_1, \ldots, x_d) \not\succeq (y_1, \ldots, y_d)$, then $C_1$ and $C_2$ are incomparable. In dimension $d$, let us therefore count the number of cells whose coordinates sum to $N + d - 1$. This corresponds to the number of integer compositions of $(N + d - 1)$ into $d$ parts, which is given by $\binom{N+d-2}{d-1}$. $\qquad\square$

The number of incomparable points, $Y_n$ is at least the number of occupied incomparable cells, which we call $\Delta$. For $d \geq 2$,

$$\mathbb{E}[\Delta] \geq \binom{N+d-2}{d-1}\left(1 - \left(1 - \frac{1}{N^d}\right)^n\right)$$

$$= \frac{(N+d-2)!}{(d-1)!(N-1)!}\left(1 - \left(1 - \frac{1}{N^d}\right)^n\right)$$

$$\geq \frac{N^{d-1}}{(d-1)!}\left(1 - \left(1 - \frac{1}{N^d}\right)^n\right)$$

Now let $N = \left\lceil n^{\frac{1}{d}} \right\rceil$. Then

$$\mathbb{E}[\Delta] \geq \frac{n^{\frac{d-1}{d}}}{(d-1)!}\left(1 - e^{-1}\right)$$

We can now lower bound the expected value of labeling number raised to the power $k$. First,

$$L(X_1, \ldots, X_n) \geq 2^{\Delta}$$
$$L(X_1, \ldots, X_n)^k \geq 2^{k\Delta}$$

By Jensen's inequality,

$$\mathbb{E}\left[L(X_1, \ldots, X_n)^k\right] \geq \mathbb{E}\left[2^{k\Delta}\right] \geq 2^{k\mathbb{E}[\Delta]} \geq 2^{k\frac{1-e^{-1}}{(d-1)!}n^{\frac{d-1}{d}}} = \exp\left[\frac{\log(2)(1-e^{-1})k}{(d-1)!}n^{\frac{d-1}{d}}\right]$$

$\qquad\square$

Finally, we tie together the above results to prove Theorem 1.

*Proof of Theorem 1.* The proof is by chaining the inequalities from Propositions 2- 5, along with Theorem 6. By Proposition 2,

$$\mathbb{P}\left(\|\hat{f}_n - f\|_2 > \epsilon\right) \leq 8N_{\mathcal{F}_{s,d}}\left(\frac{\epsilon^2}{32}, n\right)\exp\left(-\frac{\epsilon^4 n}{512}\right).$$

By Proposition 3, $N_{\mathcal{F}_{s,d}}(\epsilon, n) \leq \binom{d}{s}N_{\mathcal{F}_s}(\epsilon, n)$. Therefore,

$$\mathbb{P}\left(\|\hat{f}_n - f\|_2 > \epsilon\right) \leq 8\binom{d}{s}N_{\mathcal{F}_s}\left(\frac{\epsilon^2}{32}, n\right)\exp\left(-\frac{\epsilon^4 n}{512}\right).$$

By Proposition 4,

$$N\left(\epsilon, \mathcal{F}_s, (x_1, y_1), \ldots, (x_n, y_n)\right) \leq L\left(\left\lceil\frac{2}{\epsilon}\right\rceil, x_1, \ldots, x_n\right),$$

where $x_i \in [0, 1]^s$ for $i \in \{1, \ldots, n\}$. Substituting,

$$\mathbb{P}\left(\|\hat{f}_n - f\|_2 > \epsilon\right) \leq 8\binom{d}{s}\mathbb{E}\left[N\left(\frac{\epsilon^2}{32}, \mathcal{F}_s, (x_1, y_1), \ldots, (x_n, y_n)\right)\right]\exp\left(-\frac{\epsilon^4 n}{512}\right)$$

$$\leq 8\binom{d}{s}\mathbb{E}\left[L\left(\left\lceil\frac{64}{\epsilon^2}\right\rceil, X_1, \ldots, X_n\right)\right]\exp\left(-\frac{\epsilon^4 n}{512}\right),$$

where $X_1, \ldots, X_n$ are distributed independently and uniformly at random in $[0, 1]^s$. By Proposition 5, $L(m, x_1, \ldots, x_n) \leq (L(2, x_1, \ldots, x_n))^{m-1}$. Therefore,

$$\mathbb{P}\left(\|\hat{f}_n - f\|_2 > \epsilon\right) \leq 8\binom{d}{s}\mathbb{E}\left[L(X_1, \ldots, X_n)^{\left\lceil\frac{64}{\epsilon^2}\right\rceil - 1}\right]\exp\left(-\frac{\epsilon^4 n}{512}\right)$$

$$\leq 8\binom{d}{s}\mathbb{E}\left[L(X_1, \ldots, X_n)^{\frac{64}{\epsilon^2}}\right]\exp\left(-\frac{\epsilon^4 n}{512}\right).$$

Finally, by Theorem 6,

$$\mathbb{P}\left(\|\hat{f}_n - f\|_2 > \epsilon\right) \leq 8\binom{d}{s}\exp\left[\left(2\log(2)\frac{64}{\epsilon^2} + 2^{\frac{64}{\epsilon^2}+s}\right)n^{\frac{s-1}{s}}\right]\exp\left(-\frac{\epsilon^4 n}{512}\right)$$

$$= 8\binom{d}{s}\exp\left[\left(\frac{128\log(2)}{\epsilon^2} + 2^{\frac{64}{\epsilon^2}}2^s\right)n^{\frac{s-1}{s}} - \frac{\epsilon^4 n}{512}\right].$$

$\square$

*Proof of Corollary 1.* Equivalently, we show that $s = o\left(\sqrt{\log(n)}\right)$ and $d = e^{o\left(\frac{n}{s}\right)}$ suffices. Analyzing the leading term in the exponent,

$$2^{\frac{64}{\epsilon^2}}2^s n^{\frac{s-1}{s}} = n^{1 + \left(s + \frac{64}{\epsilon^2}\right)\frac{\log(2)}{\log(n)} - \frac{1}{s}}.$$

Analyzing the exponent,

$$1 + \left(s + \frac{64}{\epsilon^2}\right)\frac{\log(2)}{\log(n)} - \frac{1}{s} = 1 + \frac{o\left(\sqrt{\log(n)}\right)}{\log(n)} - \frac{1}{o\left(\sqrt{\log(n)}\right)}$$

$$= 1 + o\left(\frac{1}{\sqrt{\log(n)}}\right) - \omega\left(\frac{1}{\sqrt{\log(n)}}\right)$$

$$= 1 - \omega\left(\frac{1}{\sqrt{\log(n)}}\right).$$

Therefore,

$$\exp\left\{\left(\frac{128\log(2)}{\epsilon^2} + 2^{\frac{64}{\epsilon^2}}2^s\right)n^{\frac{s-1}{s}} - \frac{\epsilon^4 n}{512}\right\} = \exp\left\{\Theta(1)n^{1-\omega\left(\frac{1}{\sqrt{\log(n)}}\right)} - \Theta(n)\right\}$$

$$= \exp\left\{\Theta(n)\left(n^{-\omega\left(\frac{1}{\sqrt{\log(n)}}\right)} - 1\right)\right\}$$

$$= \exp\left\{\Theta(n)\left(e^{-\omega\left(\frac{1}{\sqrt{\log(n)}}\right)\log(n)} - 1\right)\right\}$$

$$= \exp\left\{\Theta(n)\left(e^{-\omega\left(\sqrt{\log(n)}\right)} - 1\right)\right\}$$

$$= \exp\left\{\Theta(n)\left(o\left(e^{-\sqrt{\log(n)}}\right) - 1\right)\right\}$$

$$\exp\left\{-\Theta(n)\right\}.$$

Next,

$$\binom{d}{s} \leq d^s$$

$$= e^{s\log(d)}$$

We need $s\log(d) = o(n)$, or equivalently, $d = e^{o\left(\frac{n}{s}\right)}$. $\square$

To prove Theorem 2, we first prove Lemma 2.

*Proof of Lemma 2.* Consider the following procedure. We sample $X_1$ and $X_2$ independently and uniformly on $[0,1]^d$. Fix $k \in A$. Let

$$X_+ = \begin{cases} X_1 & \text{if } X_{1,k} > X_{2,k} \\ X_2 & \text{otherwise} \end{cases}$$

and

$$X_- = \begin{cases} X_1 & \text{if } X_{1,k} \leq X_{2,k} \\ X_2 & \text{otherwise.} \end{cases}$$

In other words, $X_+$ is the right point according to coordinate $k$ and $X_-$ is the left point according to the same coordinate. Now,

$$\begin{aligned}
&\mathbb{P}\left(f(X_1) + W_1 > f(X_2) + W_2 | X_{1,k} > X_{2,k}\right) \\
&= \mathbb{P}\left(f(X_1) + W_1 > f(X_2) + W_2 | X_{1,k} = X_+, X_{2,k} = X_-\right) \\
&= \mathbb{P}\left(f(X_+) + W_1 > f(X_-) + W_2 | X_{1,k} = X_+, X_{2,k} = X_-\right)
\end{aligned}$$

We claim that the conditioning in the last expression can be dropped. Indeed,

$$\mathbb{P}\left(X_{1,k} = X_+, X_{2,k} = X_-\right) = \mathbb{P}\left(X_{1,k} = X_-, X_{2,k} = X_+\right) = \frac{1}{2}$$

so that

$$\begin{aligned}
&\mathbb{P}\left(f(X_+) + W_1 > f(X_-) + W_2\right) \\
&= \frac{1}{2}\mathbb{P}\left(f(X_+) + W_1 > f(X_-) + W_2 | X_{1,k} = X_+, X_{2,k} = X_-\right) \\
&\quad + \frac{1}{2}\mathbb{P}\left(f(X_+) + W_1 > f(X_-) + W_2 | X_{1,k} = X_-, X_{2,k} = X_+\right) \\
&= \mathbb{P}\left(f(X_+) + W_1 > f(X_-) + W_2 | X_{1,k} = X_+, X_{2,k} = X_-\right).
\end{aligned}$$

The last equality is due to the two probabilities taking the same value, by symmetry. Therefore, we have

$$\mathbb{P}\left(f(X_1) + W_1 > f(X_2) + W_2 | X_{1,k} > X_{2,k}\right) = \mathbb{P}\left(f(X_+) + W_1 > f(X_-) + W_2\right).$$

Similarly,

$$\mathbb{P}\left(f(X_1) + W_1 < f(X_2) + W_2 | X_{1,k} > X_{2,k}\right) = \mathbb{P}\left(f(X_+) + W_1 < f(X_-) + W_2\right).$$

Therefore, we can equivalently define $p_k$ as

$$p_k = \mathbb{P}\left(f(X_+) + W_1 > f(X_-) + W_2\right) - \mathbb{P}\left(f(X_+) + W_1 < f(X_-) + W_2\right).$$

Our goal is to show that

$$\mathbb{P}\left(f(X_+) + W_1 > f(X_-) + W_2\right) > \mathbb{P}\left(f(X_+) + W_1 < f(X_-) + W_2\right).$$

Let $k = 1$. By Assumption 1, the function $f$ is not constant with respect to the first coordinate.

We now construct a coupling $(\overline{X}_-, \overline{X}_+, \overline{W}_1, \overline{W}_2) \sim (X_-, X_+, W_1, W_2)$ $(\star)$ such that

$$\mathbb{P}\left(f(\overline{X}_+) + \overline{W}_1 > f(\overline{X}_-) + \overline{W}_2\right) > \mathbb{P}\left(f(X_+) + W_1 < f(X_-) + W_2\right).$$

The coupling is given by setting $\overline{X}_{+,1} = X_{+,1}$, $\overline{X}_{-,1} = X_{-,1}$. Set $\overline{X}_{+,i} = X_{-,i}$ and $\overline{X}_{-,i} = X_{+,i}$ for all $i \in \{2, \ldots, d\}$. Finally, set $\overline{W}_1 = W_2$ and $\overline{W}_2 = W_1$. Observe that by monotonicity, $f(\overline{X}_+) \geq f(X_-)$ and similarly $f(\overline{X}_-) \leq f(X_+)$. Therefore,

$$\{f(X_-) - f(X_+) \geq W_1 - W_2\} \implies \{f(\overline{X}_+) - f(\overline{X}_-) > \overline{W}_2 - \overline{W}_1\}.$$

Furthermore, $(\star)$ holds. We conclude that

$$\mathbb{P}\left(f(\overline{X}_+) + \overline{W}_1 > f(\overline{X}_-) + \overline{W}_2\right) \geq \mathbb{P}\left(f(X_+) + W_1 < f(X_-) + W_2\right).$$

To show the strict inequality, it suffices to show that there is a non-zero probability of the event

$$\{f(\overline{X}_+) + \overline{W}_1 > f(\overline{X}_-) + \overline{W}_2\} \cap \{f(X_+) + W_1 \geq f(X_-) + W_2\}$$
$$= \{f(\overline{X}_+) + W_2 > f(\overline{X}_-) + W_1\} \cap \{f(X_+) + W_1 \geq f(X_-) + W_2\}$$
$$= \{f(\overline{X}_+) - f(\overline{X}_-) > W_1 - W_2\} \cap \{f(X_+) - f(X_-) \geq W_2 - W_1\}$$
$$= \{f(\overline{X}_-) - f(\overline{X}_+) < W_2 - W_1 \leq f(X_+) - f(X_-)\}$$

This last expression holds with non-zero probability because there exists some $\epsilon > 0$ such that with positive probability, $f(\overline{X}_+) \geq f(X_-) + \epsilon$ and similarly $f(\overline{X}_-) \leq f(X_+) - \epsilon$. (Otherwise, $f$ would be constant with respect to the first coordinate).

$\square$

**Proposition 6.** *Consider stage $t$ in Algorithm 3'. Suppose that the first $t-1$ coordinates recovered by the algorithm are correct, i.e. $k_i \in \{1, \ldots, s\}$ for all $i \in \{1, \ldots, t-1\}$. Let $R = \{1, \ldots, s\} \setminus \{k_1, \ldots, k_{t-1}\}$. Let $(X_1, Y_1)$ and $(X_2, Y_2)$ be independent samples from the model. There exists $r \in R$ so that for all $k \in R$,*

$$\mathbb{P}\left(Y_1 > Y_2 | q(1,2,r) = 1, q(1,2,k) = 0\right) \geq \mathbb{P}\left(Y_2 > Y_1 | q(1,2,r) = 1, q(1,2,k) = 0\right).$$

*Proof.* Let

$$f(r,k) = \mathbb{P}\left(Y_1 > Y_2 | q(1,2,r) = 1, q(1,2,k) = 0\right) - \mathbb{P}\left(Y_2 > Y_1 | q(1,2,r) = 1, q(1,2,k) = 0\right).$$

We first claim that $f(r,k) = -f(k,r)$. Using the fact that $q(1,2,k) = 1 \iff q(2,1,k) = 0$ for all but a measure-zero set,

$$f(k,r) = \mathbb{P}\left(Y_1 > Y_2 | q(1,2,k) = 1, q(1,2,r) = 0\right) - \mathbb{P}\left(Y_2 > Y_1 | q(1,2,k) = 1, q(1,2,r) = 0\right)$$
$$= \mathbb{P}\left(Y_1 > Y_2 | q(2,1,k) = 0, q(2,1,r) = 1\right) - \mathbb{P}\left(Y_2 > Y_1 | q(2,1,k) = 0, q(2,1,r) = 1\right)$$
$$= \mathbb{P}\left(Y_2 > Y_1 | q(1,2,k) = 0, q(1,2,r) = 1\right) - \mathbb{P}\left(Y_1 > Y_2 | q(1,2,k) = 0, q(1,2,r) = 1\right)$$
$$= -f(r,k).$$

If there are one or two indices remaining to be found, then clearly such an $r$ exists. Otherwise, let $a$, $b$, and $c$ be correct indices that have not yet been found. Our next claim is that

$$\{f(a,b) \geq f(b,a), f(b,c) \geq f(c,b)\} \implies f(a,c) \geq f(c,a).$$

Suppose $f(a,b) \geq f(b,a)$ and $f(b,c) \geq f(c,b)$. Then $f(a,b) \geq 0$ and $f(b,c) \geq 0$. Observe that

$$f(a,b) \geq 0$$
$$\iff \mathbb{P}\left(Y_1 > Y_2 | q(1,2,a) = 1, q(1,2,b) = 0\right) - \mathbb{P}\left(Y_2 > Y_1 | q(1,2,a) = 1, q(1,2,b) = 0\right) \geq 0$$
$$\iff \mathbb{P}\left(Y_1 > Y_2, q(1,2,a) = 1, q(1,2,b) = 0\right) - \mathbb{P}\left(Y_2 > Y_1, q(1,2,a) = 1, q(1,2,b) = 0\right) \geq 0$$

Similarly,

$$f(b,c) \geq 0$$
$$\iff \mathbb{P}\left(Y_1 > Y_2, q(1,2,b) = 1, q(1,2,c) = 0\right) - \mathbb{P}\left(Y_2 > Y_1, q(1,2,b) = 1, q(1,2,c) = 0\right) \geq 0$$

and

$$f(a,c) \geq 0$$
$$\iff \mathbb{P}\left(Y_1 > Y_2, q(1,2,a) = 1, q(1,2,c) = 0\right) - \mathbb{P}\left(Y_2 > Y_1, q(1,2,a) = 1, q(1,2,c) = 0\right) \geq 0.$$

By the Law of Total Probability,

$$\mathbb{P}\left(Y_1 > Y_2, q(1,2,a) = 1, q(1,2,c) = 0\right)$$
$$= \mathbb{P}\left(Y_1 > Y_2, q(1,2,a) = 1, q(1,2,b) = 1, q(1,2,c) = 0\right)$$
$$+ \mathbb{P}\left(Y_1 > Y_2, q(1,2,a) = 1, q(1,2,b) = 0, q(1,2,c) = 0\right). \tag{24}$$

Consider the first term of Equation (24).

$$\mathbb{P}\left(Y_1 > Y_2, q(1,2,a) = 1, q(1,2,b) = 1, q(1,2,c) = 0\right)$$
$$= \mathbb{P}\left(Y_1 > Y_2, q(1,2,b) = 1, q(1,2,c) = 0\right) - \mathbb{P}\left(Y_1 > Y_2, q(1,2,a) = 0, q(1,2,b) = 1, q(1,2,c) = 0\right)$$

Similarly, consider the second term of Equation (24).

$$\mathbb{P}\left(Y_1 > Y_2, q(1,2,a) = 1, q(1,2,b) = 0, q(1,2,c) = 0\right)$$
$$= \mathbb{P}\left(Y_1 > Y_2, q(1,2,a) = 1, q(1,2,b) = 0\right) - \mathbb{P}\left(Y_1 > Y_2, q(1,2,a) = 1, q(1,2,b) = 0, q(1,2,c) = 1\right)$$

Adding the terms,

$$\mathbb{P}\left(Y_1 > Y_2, q(1,2,a) = 1, q(1,2,c) = 0\right)$$
$$= \mathbb{P}\left(Y_1 > Y_2, q(1,2,b) = 1, q(1,2,c) = 0\right) - \mathbb{P}\left(Y_1 > Y_2, q(1,2,a) = 0, q(1,2,b) = 1, q(1,2,c) = 0\right)$$
$$+ \mathbb{P}\left(Y_1 > Y_2, q(1,2,a) = 1, q(1,2,b) = 0\right) - \mathbb{P}\left(Y_1 > Y_2, q(1,2,a) = 1, q(1,2,b) = 0, q(1,2,c) = 1\right).$$

Analyzing the expression $\mathbb{P}\left(Y_2 > Y_1, q(1,2,a) = 1, q(1,2,c) = 0\right)$ similarly, we obtain

$$\mathbb{P}\left(Y_2 > Y_1, q(1,2,a) = 1, q(1,2,c) = 0\right)$$
$$= \mathbb{P}\left(Y_2 > Y_1, q(1,2,b) = 1, q(1,2,c) = 0\right) - \mathbb{P}\left(Y_2 > Y_1, q(1,2,a) = 0, q(1,2,b) = 1, q(1,2,c) = 0\right)$$
$$+ \mathbb{P}\left(Y_2 > Y_1, q(1,2,a) = 1, q(1,2,b) = 0\right) - \mathbb{P}\left(Y_2 > Y_1, q(1,2,a) = 1, q(1,2,b) = 0, q(1,2,c) = 1\right).$$

Recall that we need to show

$$\mathbb{P}\left(Y_1 > Y_2, q(1,2,a) = 1, q(1,2,c) = 0\right) - \mathbb{P}\left(Y_2 > Y_1, q(1,2,a) = 1, q(1,2,c) = 0\right) \geq 0.$$

Taking the difference,

$$\mathbb{P}\left(Y_1 > Y_2, q(1,2,a) = 1, q(1,2,c) = 0\right) - \mathbb{P}\left(Y_2 > Y_1, q(1,2,a) = 1, q(1,2,c) = 0\right)$$
$$= f(a,b) + f(b,c)$$
$$\quad - \mathbb{P}\left(Y_1 > Y_2, q(1,2,a) = 0, q(1,2,b) = 1, q(1,2,c) = 0\right)$$
$$\quad - \mathbb{P}\left(Y_1 > Y_2, q(1,2,a) = 1, q(1,2,b) = 0, q(1,2,c) = 1\right)$$
$$\quad + \mathbb{P}\left(Y_2 > Y_1, q(1,2,a) = 0, q(1,2,b) = 1, q(1,2,c) = 0\right)$$
$$\quad + \mathbb{P}\left(Y_2 > Y_1, q(1,2,a) = 1, q(1,2,b) = 0, q(1,2,c) = 1\right)$$
$$= f(a,b) + f(b,c)$$
$$\quad - \mathbb{P}\left(Y_1 > Y_2, q(1,2,a) = 0, q(1,2,b) = 1, q(1,2,c) = 0\right)$$
$$\quad - \mathbb{P}\left(Y_1 > Y_2, q(1,2,a) = 1, q(1,2,b) = 0, q(1,2,c) = 1\right)$$
$$\quad + \mathbb{P}\left(Y_2 > Y_1, q(2,1,a) = 1, q(2,1,b) = 0, q(2,1,c) = 1\right)$$
$$\quad + \mathbb{P}\left(Y_2 > Y_1, q(2,1,a) = 0, q(2,1,b) = 1, q(2,1,c) = 0\right)$$
$$= f(a,b) + f(b,c)$$
$$\geq 0.$$

Therefore, we have proven the claim

$$\{f(a,b) \geq f(b,a), f(b,c) \geq f(c,b)\} \implies f(a,c) \geq f(c,a).$$

This fact shows that the indices can be totally ordered, i.e. by writing $a \geq b$ when $f(a,b) \geq f(b,a)$. We let $r$ be the largest element of the order. $\square$

**Lemma 7.** *Consider stage $t$ in Algorithm 3', which uses $m = \frac{n}{s}$ samples. The probability that the first coordinate is correctly recovered is at least*

$$1 - (d-1)\exp\left(-\frac{p_1^2 m}{64 s^2}\right).$$

*Suppose that the first $t-1$ coordinates recovered by the algorithm are correct, i.e. $k_i \in \{1,\ldots,s\}$ for all $i \in \{1,\ldots,t-1\}$. Then $k_t \in \{1,\ldots,s\}$ with probability at least*

$$1 - (d-t)\exp\left(-\frac{p_1^2 m}{64(s-t+1)^2}\right).$$

*Proof.* Applying Proposition 6, let $r$ be an element of $\{1,\ldots,s\} \setminus \{k_1,\ldots,k_{t-1}\}$ such that

$$\mathbb{P}\left(Y_1 > Y_2 | q(1,2,r) = 1, q(1,2,k) = 0\right) \geq \mathbb{P}\left(Y_2 > Y_1 | q(1,2,r) = 1, q(1,2,k) = 0\right).$$

for all $k \in \{1,\ldots,s\} \setminus \{k_1,\ldots,k_{t-1}\}$.

Next, we consider the optimization problem at step $t$. Fix a feasible solution $v = \overline{v}$. Recall that $\sum_{k=1}^{d} \overline{v}_k = 1$. For this fixed value $\overline{v}$, the optimal choice for the variables $c_k^{ij}$ must satisfy

$$\sum_{k=1}^{d} q(i,j,k) c_k^{ij} = 1 - \sum_{k=1}^{d} q(i,j,k) \overline{v}_k$$

for $i, j$ such that $Y_i > Y_j$ and $\sum_{k=1}^{d} q(i,j,k) \geq 1$, with $c_k^{ij} = 0$ whenever $q(i,j,k) = 0$. Note that $\sum_{k=1}^{d} q(i,j,k) c_k^{ij} = \sum_{k=1}^{d} c_k^{ij}$. Therefore, the objective function is equal to

$$z(\overline{v}) \triangleq \sum_{i=1}^{m} \sum_{j=1}^{m} \mathbb{1}\{Y_i > Y_j, X_i \not\preceq X_j\} \left(1 - \sum_{k=1}^{d} q(i,j,k) \overline{v}_k\right).$$

Let $v^\star = e_r$. Let $F_t$ be the feasible set for the vector of variables $v$ at step $t$, i.e.

$$F_t = \left\{ v \in \mathbb{R}^d : v_i \geq 0 \; \forall i \in \{1, \ldots, d\}, v_i = 0 \; \forall i \in \{k_1, \ldots, k_{t-1}\}, \sum_{i=1}^{d} v_i = 1 \right\}.$$

Let $\overline{F}_t = \{v \in F_t : \arg\max_i v_i \cap \{s+1, \ldots, d\} \neq \emptyset\}$. In other words, $\overline{F}_t$ is the set of feasible solutions that lead to an incorrect coordinate choice at step $t$. We will give an upper bound on the probability

$$\mathbb{P}\left(\exists v \in \overline{F}_t : z(v) \leq z(v^\star)\right).$$

Note that the complementary event, $\{z(v) > z(v^\star), \forall v \in \overline{F}_t\}$, implies that the optimization problem will choose a coordinate among $\{1, \ldots, s\} \setminus \{k_1, \ldots, k_{t-1}\}$.

Let $\overline{v} \in \overline{F}_t$ and write $\overline{v} = v^\star + u$. Observe that since the coordinates of $\overline{v}$ and $v^\star$ both sum to 1, the coordinates of $u$ sum to 0. Also, $u_r < 0$ and $u_k \geq 0$ for $k \neq r$. Now,

$$z(\overline{v}) - z(v^\star)$$

$$= \sum_{i=1}^{m} \sum_{j=1}^{m} \mathbb{1}\{Y_i > Y_j, X_i \not\preceq X_j\} \left(\left(1 - \sum_{k=1}^{d} q(i,j,k)\overline{v}_k\right) - \left(1 - \sum_{k=1}^{d} q(i,j,k)v_k^\star\right)\right)$$

$$= \sum_{i=1}^{m} \sum_{j=1}^{m} \mathbb{1}\{Y_i > Y_j, X_i \not\preceq X_j\} \left(\sum_{k=1}^{d} q(i,j,k)v_k^\star - \sum_{k=1}^{d} q(i,j,k)\overline{v}_k\right)$$

$$= \sum_{i=1}^{m} \sum_{j=1}^{m} \mathbb{1}\{Y_i > Y_j, X_i \not\preceq X_j\} \sum_{k=1}^{d} q(i,j,k) \left(v_k^\star - \overline{v}_k\right)$$

$$= -\sum_{i=1}^{m} \sum_{j=1}^{m} \mathbb{1}\{Y_i > Y_j, X_i \not\preceq X_j\} \sum_{k=1}^{d} q(i,j,k) u_k$$

$$= -\sum_{k=1}^{d} u_k \sum_{i=1}^{m} \sum_{j=1}^{m} \mathbb{1}\{Y_i > Y_j, X_i \not\preceq X_j\} q(i,j,k).$$

Now,

$$\mathbb{1}\{Y_i > Y_j, X_i \not\preceq X_j\} q(i,j,k) = \mathbb{1}\{Y_i > Y_j\} \mathbb{1}\{X_i \not\preceq X_j\} q(i,j,k)$$
$$= \mathbb{1}\{Y_i > Y_j\} (1 - \mathbb{1}\{X_i \preceq X_j\}) q(i,j,k)$$
$$= \mathbb{1}\{Y_i > Y_j\} q(i,j,k) - \mathbb{1}\{Y_i > Y_j\} \mathbb{1}\{X_i \preceq X_j\} q(i,j,k)$$
$$= \mathbb{1}\{Y_i > Y_j\} q(i,j,k),$$

where the last equality is due to the fact that $X_i \preceq X_j$ implies $q(i,j,k) = 0$. Substituting,

$$z(\overline{v}) - z(v^\star) = -\sum_{k=1}^{d} u_k \sum_{i=1}^{m} \sum_{j=1}^{m} \mathbb{1}\{Y_i > Y_j\} q(i,j,k)$$

$$= -u_r \sum_{i=1}^m \sum_{j=1}^m \mathbb{1}\{Y_i > Y_j\} q(i,j,r) - \sum_{k \neq r} u_k \sum_{i=1}^n \sum_{j=1}^n \mathbb{1}\{Y_i > Y_j\} q(i,j,k)$$

$$= \sum_{k \neq r} u_k \sum_{i=1}^m \sum_{j=1}^m \mathbb{1}\{Y_i > Y_j\} q(i,j,r) - \sum_{k \neq r} u_k \sum_{i=1}^n \sum_{j=1}^n \mathbb{1}\{Y_i > Y_j\} q(i,j,k)$$

$$= \sum_{k \neq r} u_k \sum_{i=1}^m \sum_{j=1}^m \mathbb{1}\{Y_i > Y_j\} \left( q(i,j,r) - q(i,j,k) \right).$$

Recall that we seek to upper bound the probability $\mathbb{P}\left( \exists v \in \overline{F}_t : z(v) \leq z(v^\star) \right)$. From the above,

$$\mathbb{P}\left( \exists v \in \overline{F}_t : z(v) \leq z(v^\star) \right) = \mathbb{P}\left( \exists v^\star + u \in \overline{F}_t : z(v^\star + u) \leq z(v^\star) \right)$$

$$= \mathbb{P}\left( \exists v^\star + u \in \overline{F}_t : \sum_{k \neq r} u_k \sum_{i=1}^m \sum_{j=1}^m \mathbb{1}\{Y_i > Y_j\} \left( q(i,j,r) - q(i,j,k) \right) \leq 0 \right).$$

Consider $v^\star + u \in \overline{F}_t$. Recalling that $u_k = 0$ for all $k \in \{k_1, \ldots, k_{t-1}\}$, observe that

$$\sum_{k \in \{s+1,\ldots,d\}} u_k \geq \frac{1}{s-t} \sum_{k \in \{1,\ldots,s\} \setminus \{r\}} u_k$$

$$\iff (s-t) \sum_{k \in \{s+1,\ldots,d\}} u_k \geq -u_r - \sum_{k \in \{s+1,\ldots,d\}} u_k$$

$$\iff \sum_{k \in \{s+1,\ldots,d\}} u_k \geq \frac{1}{s-t+1}(-u_r). \tag{25}$$

Since $-u_r > 0$,

$$\mathbb{P}\left( \exists v^\star + u \in \overline{F}_t : \sum_{k \neq r} u_k \sum_{i=1}^m \sum_{j=1}^m \mathbb{1}\{Y_i > Y_j\} \left( q(i,j,r) - q(i,j,k) \right) \leq 0 \right)$$

$$= \mathbb{P}\left( \exists v^\star + u \in \overline{F}_t : \frac{1}{-u_r} \sum_{k \neq r} u_k \sum_{i=1}^m \sum_{j=1}^m \mathbb{1}\{Y_i > Y_j\} \left( q(i,j,r) - q(i,j,k) \right) \leq 0 \right).$$

Let $0 < \Delta \leq \frac{p_r}{4(s-t+1)}$. Observe that the existence of $v^\star + u \in \overline{F}_t$ such that $\frac{1}{-u_r} \sum_{k \neq r} u_k \sum_{i=1}^m \sum_{j=1}^m \mathbb{1}\{Y_i > Y_j\} \left( q(i,j,r) - q(i,j,k) \right) \leq 0$ implies at least one of the following occurs:

1. There exists $k \in \{1, \ldots, s\} \setminus \{r, k_1, \ldots, k_{t-1}\}$ such that

$$\sum_{i=1}^m \sum_{j=1}^m \mathbb{1}\{Y_i > Y_j\} \left( q(i,j,r) - q(i,j,k) \right) \leq m(m-1)(-\Delta).$$

2. There exists $k \in \{s+1, \ldots, d\}$ such that

$$\sum_{i=1}^m \sum_{j=1}^m \mathbb{1}\{Y_i > Y_j\} \left( q(i,j,r) - q(i,j,k) \right) \leq m(m-1)\left( \frac{1}{4}p_r - \Delta \right).$$

Indeed, if none of these events occur, then for every $v^\star + u \in \overline{F}_t$,

$$\frac{1}{-u_r} \sum_{k \neq r} u_k \sum_{i=1}^m \sum_{j=1}^m \mathbb{1}\{Y_i > Y_j\} \left( q(i,j,r) - q(i,j,k) \right)$$

$$= \frac{1}{-u_r} \sum_{k \in \{1,\ldots,s\} \setminus \{r, k_1, \ldots, k_{t-1}\}} u_k \sum_{i=1}^m \sum_{j=1}^m \mathbb{1}\{Y_i > Y_j\} \left( q(i,j,r) - q(i,j,k) \right)$$

$$+ \frac{1}{-u_r} \sum_{k \in \{s+1,\ldots,d\}} u_k \sum_{i=1}^{m} \sum_{j=1}^{m} \mathbb{1}\{Y_i > Y_j\}\left(q(i,j,r) - q(i,j,k)\right)$$

$$> m(m-1)\left[\frac{1}{-u_r} \sum_{k \in \{1,\ldots,s\}\setminus\{r,k_1,\ldots,k_{t-1}\}} u_k(-\Delta) + \frac{1}{-u_r} \sum_{k \in \{s+1,\ldots,d\}} u_k\left(\frac{1}{4}p_r - \Delta\right)\right]$$

$$= m(m-1)\left[\frac{-\Delta}{-u_r}\sum_{k \neq r} u_k + \frac{p_r}{-4u_r}\sum_{k \in \{s+1,\ldots,d\}} u_k\right]$$

$$= m(m-1)\left[-\Delta + \frac{p_r}{-4u_r}\sum_{k \in \{s+1,\ldots,d\}} u_k\right]$$

$$\geq m(m-1)\left[-\Delta + \frac{p_r}{4(s-t+1)}\right] \tag{26}$$

$$\geq m(m-1)\left[-\frac{p_r}{4(s-t+1)} + \frac{p_r}{4(s-t+1)}\right]$$

$$= 0.$$

The inequality (26) holds by (25). Therefore, by the Union Bound,

$$\mathbb{P}\left(\exists v^\star + u \in \overline{F}_t : \frac{1}{-u_r}\sum_{k \neq r} u_k \sum_{i=1}^{m}\sum_{j=1}^{m}\mathbb{1}\{Y_i > Y_j\}\left(q(i,j,r) - q(i,j,k)\right) \leq 0\right)$$

$$\leq \sum_{k \in \{1,\ldots,s\}\setminus\{r,k_1,\ldots,k_{t-1}\}} \mathbb{P}\left(\sum_{i=1}^{m}\sum_{j=1}^{m}\mathbb{1}\{Y_i > Y_j\}\left(q(i,j,r) - q(i,j,k)\right) \leq -m(m-1)\Delta\right)$$

$$+ \sum_{k \in \{s+1,\ldots,d\}} \mathbb{P}\left(\sum_{i=1}^{m}\sum_{j=1}^{m}\mathbb{1}\{Y_i > Y_j\}\left(q(i,j,r) - q(i,j,k)\right) \leq m(m-1)\left(\frac{1}{4}p_r - \Delta\right)\right)$$

We upper bound each probability by establishing concentration. Fix $i, j \in \{1,\ldots,m\}$ with $i \neq j$, and $k \neq r$. By the Law of Total Expectation,

$$\mathbb{E}\left[\mathbb{1}\{Y_i > Y_j\}\left(q(i,j,r) - q(i,j,k)\right)\right]$$
$$= \mathbb{E}\left[\mathbb{1}\{Y_i > Y_j\}\,|q(i,j,r) = 1, q(i,j,k) = 0\right]\mathbb{P}\left(q(i,j,r) = 1, q(i,j,k) = 0\right)$$
$$- \mathbb{E}\left[\mathbb{1}\{Y_i > Y_j\}\,|q(i,j,r) = 0, q(i,j,k) = 1\right]\mathbb{P}\left(q(i,j,r) = 0, q(i,j,k) = 1\right)$$
$$= \mathbb{P}\left(Y_i > Y_j|q(i,j,r) = 1, q(i,j,k) = 0\right)\mathbb{P}\left(q(i,j,r) = 1, q(i,j,k) = 0\right)$$
$$- \mathbb{P}\left(Y_i > Y_j|q(i,j,r) = 0, q(i,j,k) = 1\right)\mathbb{P}\left(q(i,j,r) = 0, q(i,j,k) = 1\right)$$
$$= \mathbb{P}\left(Y_i > Y_j, q(i,j,r) = 1, q(i,j,k) = 0\right) - \mathbb{P}\left(Y_i > Y_j, q(i,j,r) = 0, q(i,j,k) = 1\right).$$

Now, $q(i,j,k) = 1 \iff q(j,i,k) = 0$. Therefore,

$$\mathbb{E}\left[\mathbb{1}\{Y_i > Y_j\}\left(q(i,j,r) - q(i,j,k)\right)\right]$$
$$= \mathbb{P}\left(Y_i > Y_j, q(i,j,r) = 1, q(i,j,k) = 0\right) - \mathbb{P}\left(Y_i > Y_j, q(j,i,r) = 1, q(j,i,k) = 0\right)$$
$$= \mathbb{P}\left(Y_i > Y_j, q(i,j,r) = 1, q(i,j,k) = 0\right) - \mathbb{P}\left(Y_j > Y_i, q(i,j,r) = 1, q(i,j,k) = 0\right)$$
$$= \left[\mathbb{P}\left(Y_i > Y_j|q(i,j,r) = 1, q(i,j,k) = 0\right) - \mathbb{P}\left(Y_j > Y_i|q(i,j,r) = 1, q(i,j,k) = 0\right)\right]\mathbb{P}\left(q(i,j,r) = 1, q(i,j,k) = 0\right)$$
$$= \frac{1}{4}\left[\mathbb{P}\left(Y_i > Y_j|q(i,j,r) = 1, q(i,j,k) = 0\right) - \mathbb{P}\left(Y_j > Y_i|q(i,j,r) = 1, q(i,j,k) = 0\right)\right],$$

where we have swapped $i$ and $j$ in the second equality, due to symmetry. We now consider the two cases for $k$. First consider $k \in \{1,\ldots,s\}\setminus\{r,k_1,\ldots,k_{t-1}\}$. Due to the choice of $r$, the expectation is nonnegative, and we lower bound it by $0$.

Next consider $k \in \{s+1,\ldots,d\}$. Due to the independence of the coordinates of $X$, the values of the non-active coordinates does not influence the value of the active coordinates. Also, the value of

the function is determined entirely by the active coordinates. Therefore, we can drop the conditioning on the ordering on the inactive coordinate $k$.

For $k \in \{s+1, \dots, d\}$, we therefore have

$$\mathbb{E}\left[\mathbb{1}\{Y_i > Y_j\}(q(i,j,r) - q(i,j,k))\right] = \frac{1}{4}\left[\mathbb{P}(Y_i > Y_j | q(i,j,r) = 1) - \mathbb{P}(Y_j > Y_i | q(i,j,r) = 1)\right]$$

$$= \frac{1}{4}p_r.$$

Let $k \in \{1, \dots, s\} \setminus \{r, k_1, \dots, k_{t-1}\}$. Since $\mathbb{E}\left[\sum_{i=1}^{m}\sum_{j=1}^{m}\mathbb{1}\{Y_i > Y_j\}(q(i,j,r) - q(i,j,k))\right] \geq 0$, we have

$$\mathbb{P}\left(\sum_{i=1}^{m}\sum_{j=1}^{m}\mathbb{1}\{Y_i > Y_j\}(q(i,j,r) - q(i,j,k)) \leq -m(m-1)\Delta\right)$$

$$\leq \mathbb{P}\left(\sum_{i=1}^{m}\sum_{j=1}^{m}\mathbb{1}\{Y_i > Y_j\}(q(i,j,r) - q(i,j,k)) \leq \mathbb{E}\left[\sum_{i=1}^{m}\sum_{j=1}^{m}\mathbb{1}\{Y_i > Y_j\}(q(i,j,r) - q(i,j,k))\right] - m(m-1)\Delta\right)$$

Similarly for $k \in \{s+1, \dots, d\}$, we have

$$\mathbb{P}\left(\sum_{i=1}^{m}\sum_{j=1}^{m}\mathbb{1}\{Y_i > Y_j\}(q(i,j,r) - q(i,j,k)) \leq m(m-1)\left(\frac{1}{4}p_r - \Delta\right)\right)$$

$$= \mathbb{P}\left(\sum_{i=1}^{m}\sum_{j=1}^{m}\mathbb{1}\{Y_i > Y_j\}(q(i,j,r) - q(i,j,k)) \leq \mathbb{E}\left[\sum_{i=1}^{m}\sum_{j=1}^{m}\mathbb{1}\{Y_i > Y_j\}(q(i,j,r) - q(i,j,k))\right] - m(m-1)\Delta\right)$$

Consider the summation $\sum_{i=1}^{m}\sum_{j=1}^{m}\mathbb{1}\{Y_i > Y_j\}(q(i,j,r) - q(i,j,k))$, as a function of the $X_i$ and $W_i$ variables, for fixed $k$. We now establish the bounded differences property for the $X_i$ and $W_i$ variables. Suppose we change the value of $W_i$. The affected terms are $\mathbb{1}\{Y_i > Y_j\}(q(i,j,r) - q(i,j,k))$ and $\mathbb{1}\{Y_j > Y_i\}(q(j,i,r) - q(j,i,k))$, for all $j \neq i$. Fix $j \neq i$. The largest absolute change is 2, and occurs when $q(i,j,r) = 1$, $q(i,j,k) = 0$, and $Y_i > Y_j$, and changing $W_i$ switches the order on $Y_i$ and $Y_j$. Adding the contributions for all $j \neq i$, the total change corresponding to changing $W_i$ is bounded by $2(m-1)$. By similar reasoning, changing any $X_i$ may change the summation by up to $2(m-1)$.

Applying the McDiarmid inequality, we obtain for every $k \notin \{r, k_1, \dots, k_{t-1}\}$,

$$\mathbb{P}\left(\sum_{i=1}^{m}\sum_{j=1}^{m}\mathbb{1}\{Y_i > Y_j\}(q(i,j,r) - q(i,j,k)) \leq \mathbb{E}\left[\sum_{i=1}^{m}\sum_{j=1}^{m}\mathbb{1}\{Y_i > Y_j\}(q(i,j,r) - q(i,j,k))\right] - m(m-1)\Delta\right)$$

$$\leq \exp\left(-\frac{2(\Delta m(m-1))^2}{2m(2(m-1))^2}\right)$$

$$= \exp\left(-\frac{\Delta^2 m^2(m-1)^2}{4m(m-1)^2}\right)$$

$$= \exp\left(-\frac{1}{4}\Delta^2 m\right).$$

Substituting $\Delta = \frac{p_r}{4(s-t+1)}$, we obtain

$$\mathbb{P}\left(\sum_{i=1}^{m}\sum_{j=1}^{m}\mathbb{1}\{Y_i > Y_j\}(q(i,j,r) - q(i,j,k)) \leq \mathbb{E}\left[\sum_{i=1}^{m}\sum_{j=1}^{m}\mathbb{1}\{Y_i > Y_j\}(q(i,j,r) - q(i,j,k))\right] - m(m-1)\Delta\right)$$

$$\leq \exp\left(-\frac{p_r^2 m}{64(s-t+1)^2}\right)$$

Finally,

$$\mathbb{P}\left(\exists v^{\star} + u \in \overline{F}_t : \frac{1}{-u_r} \sum_{k \neq r} u_k \sum_{i=1}^{m} \sum_{j=1}^{m} \mathbb{1}\{Y_i > Y_j\}\left(q(i,j,r) - q(i,j,k)\right) \leq 0\right) \leq (d-t)\exp\left(-\frac{p_r^2 m}{64(s-t+1)^2}\right).$$

We conclude that the probability that the $t$th coordinate is correct is lower-bounded by

$$1 - (d-t)\exp\left(-\frac{p_r^2 m}{64(s-t+1)^2}\right) \geq 1 - (d-t)\exp\left(-\frac{p_1^2 m}{64(s-t+1)^2}\right).$$

$\square$

*Proof of Theorem 2.* Let $k_t$ be the $t$th coordinate recovered, where $t \in \{1, \ldots, s\}$. By Lemma 7, the probability that $k_1 \notin \{1, \ldots, s\}$ is upper bounded by $(d-1)\exp\left(-\frac{p_1^2 m}{64s^2}\right)$.. Next, the probability that $k_2 \notin \{1, \ldots, s\}$ given that $k_1 \in \{1, \ldots, s\}$ is upper bounded by $(d-2)\exp\left(-\frac{p_1^2 m}{64(s-1)^2}\right)$. In general, the probability that $k_t \notin \{1, \ldots, s\}$ given that $k_i \in \{1, \ldots, s\}$ for all $i \in \{1, \ldots, t-1\}$ is at most $(d-t)\exp\left(-\frac{p_1^2 m}{64(s-t+1)^2}\right)$. Therefore, the probability of error in any coordinate is upper bounded by

$$\sum_{t=1}^{s}(d-t)\exp\left(-\frac{p_1^2 m}{64(s-t+1)^2}\right) \leq \sum_{t=1}^{s}(d-t)\exp\left(-\frac{p_1^2 m}{64s^2}\right)$$

$$\leq ds\exp\left(-\frac{p_1^2 n}{64s^3}\right).$$

$\square$

*Proof of Corollary 2.*

$$ds\exp\left(-\frac{p_1^2 n}{64s^3}\right) = \exp\left(\log(d) + \log(s) - \frac{p_1^2 n}{64s^3}\right)$$

$$\leq \exp\left(2\log(d) - \frac{p_1^2 n}{64s^3}\right) \qquad (27)$$

Therefore, if $n = \omega(s^3 \log(d))$, then (27) goes to zero. $\square$

*Proof of Corollary 3.* Support recovery fails with probability at most

$$ds\exp\left(-\frac{p_1^2 n}{64s^3}\right).$$

If it succeeds, the probability of the $L_2$ norm error exceeding $\epsilon$ is upper bounded by the value in Theorem 1, with $d$ set to $s$ and $n$ set to $m = \frac{n}{s}$. Then $\mathbb{P}\left(\|\hat{f}_n - f\|_2^2 > \epsilon\right)$ is at most

$$ds\exp\left(-\frac{p_1^2 n}{64s^3}\right) + 6\exp\left[\left(\frac{2^{12}}{\epsilon^2}\log(2) + 2^{\frac{2^{11}}{\epsilon^2}}2^s\right)m^{\frac{s-1}{s}} - \frac{3\epsilon^3 m}{41 \times 2^{10}}\right].$$

Therefore, if $n = \omega(s^3\log(d))$ and $n = se^{\omega(s^2)}$, the estimator is consistent, by Corollaries 1 and 2. $\square$

## C  Proofs for the Noisy Input Model

*Proof of Theorem 3.* To illustrate the proof idea, we show the claim for $d = s = 1$ first. Observe that for any monotone partition $(S_0, S_1)$ in $\mathbb{R}$, either $S_0 = \{x : x \leq r\}$ or $S_0 = \{x : x < r\}$ for some $r$. When $d = s = 1$, the optimization problem (5)-(9) amounts to finding a boundary $r \in \mathbb{R}$. Let

$$g\left(X_{1:n}, W_{1:n}; (S_0, S_1)\right) = \sum_{i=1}^{n} \mathbb{1}\{f(X_i + W_i) = 1, X_i \in S_0\} + \mathbb{1}\{f(X_i + W_i) = 0, X_i \in S_1\}$$

denote the corresponding value of the objective function. Observe that the value of $g\left(X_{1:n}, W_{1:n};\left(S_0, S_1\right)\right)$ can change by at most $\pm 2$ when any one of the random variables is changed. Applying the McDiarmid inequality, for all $\epsilon > 0$, it holds that

$$\mathbb{P}\left(g\left(X_{1:n}, W_{1:n};\left(S_0, S_1\right)\right) - \mathbb{E}\left[g\left(X_{1:n}, W_{1:n};\left(S_0, S_1\right)\right)\right] \geq \epsilon n\right) \leq \exp\left(-\frac{2\epsilon^2 n^2}{2n \cdot 2^2}\right)$$

$$= \exp\left(-\frac{\epsilon^2 n}{4}\right).$$

Similarly,

$$\mathbb{P}\left(g\left(X_{1:n}, W_{1:n};\left(S_0, S_1\right)\right) - \mathbb{E}\left[g\left(X_{1:n}, W_{1:n};\left(S_0, S_1\right)\right)\right] \leq -\epsilon n\right) \leq \exp\left(-\frac{\epsilon^2 n}{4}\right).$$

We now calculate $\mathbb{E}\left[g\left(X_{1:n}, W_{1:n};\left(S_0, S_1\right)\right)\right]$:

$$\mathbb{E}\left[g\left(X_{1:n}, W_{1:n};\left(S_0, S_1\right)\right)\right] = n\left[p\int_{t\in S_1} h_0(t)dt + (1-p)\int_{t\in S_0} h_1(t)dt\right]$$

$$= n\left[pH_0(S_1) + (1-p)H_1(S_0)\right]$$

$$= n \cdot q(S_0, S_1).$$

By Assumption 2, the expectation has a unique minimizer $(S_0^\star, S_1^\star) \in \mathcal{M}_1$.

Observe that

$$\mathbb{P}\left(\|\hat{f}_n - f\|_2^2 > \delta\right) = \mathbb{P}\left(D\left((S_0, S_1), (S_0^\star, S_1^\star)\right) > \delta\right)$$

$$= \mathbb{P}\left((S_0, S_1) \notin B_\delta(S_0^\star, S_1^\star)\right)$$

We therefore need to analyze the probability that there exists a monotone partition outside $B_\delta(S_0^\star, S_1^\star)$ with a smaller value of $g$ than $g\left(X_{1:n}, W_{1:n};\left(S_0^\star, S_1^\star\right)\right)$. For all $(S_0, S_1) \in \mathcal{M}_1$,

$$\mathbb{E}\left[g\left(X_{1:n}, W_{1:n};\left(S_0, S_1\right)\right)\right] - \mathbb{E}\left[g\left(X_{1:n}, W_{1:n};\left(S_0^\star, S_1^\star\right)\right)\right] = n\left(q(S_0, S_1) - q\left(S_0^\star, S_1^\star\right)\right)$$

We now use the concentration result with $\epsilon$ set to $\frac{1}{3}\left(q(S_0, S_1) - q\left(S_0^\star, S_1^\star\right)\right)$. For any $(S_0, S_1)$, with probability at least

$$1 - \exp\left(-\frac{\left(q(S_0, S_1) - q\left(S_0^\star, S_1^\star\right)\right)^2 n}{36}\right),$$

it holds that

$$g\left(X_{1:n}, W_{1:n};\left(S_0, S_1\right)\right) \geq \mathbb{E}\left[g\left(X_{1:n}, W_{1:n};\left(S_0, S_1\right)\right)\right] - \frac{n}{3}\left(q(S_0, S_1) - q\left(S_0^\star, S_1^\star\right)\right).$$

Similarly, with the same probability, it holds that

$$g\left(X_{1:n}, W_{1:n};\left(S_0^\star, S_1^\star\right)\right) \leq \mathbb{E}\left[g\left(X_{1:n}, W_{1:n};\left(S_0^\star, S_1^\star\right)\right)\right] + \frac{n}{3}\left(q(S_0, S_1) - q\left(S_0^\star, S_1^\star\right)\right).$$

For a given $(S_0, S_1) \neq (S_0^\star, S_1^\star)$, both of these events occur with probability at least

$$1 - 2\exp\left(-\frac{\left(q(S_0, S_1) - q\left(S_0^\star, S_1^\star\right)\right)^2 n}{36}\right).$$

In that case,

$$g\left(X_{1:n}, W_{1:n};\left(S_0, S_1\right)\right) - g\left(X_{1:n}, W_{1:n};\left(S_0^\star, S_1^\star\right)\right)$$

$$\geq \mathbb{E}\left[g\left(X_{1:n}, W_{1:n};\left(S_0, S_1\right)\right)\right] - \frac{n}{3}\left(q(S_0, S_1) - q\left(S_0^\star, S_1^\star\right)\right)$$

$$- \mathbb{E}\left[g\left(X_{1:n}, W_{1:n};\left(S_0^\star, S_1^\star\right)\right)\right] - \frac{n}{3}\left(q(S_0, S_1) - q\left(S_0^\star, S_1^\star\right)\right)$$

$$= n\left(q(S_0, S_1) - q\left(S_0^\star, S_1^\star\right)\right) - \frac{2n}{3}\left(q(S_0, S_1) - q\left(S_0^\star, S_1^\star\right)\right)$$

$$= \frac{n}{3}\left(q(S_0, S_1) - q\left(S_0^\star, S_1^\star\right)\right).$$

Therefore, in this situation, solution $(S_0, S_1)$ is suboptimal compared to solution $(S_0^\star, S_1^\star)$.

Observe that the cardinality of the set $\{g(X_{1:n}, W_{1:n}; (S_0, S_1)) : (S_0, S_1) \in \mathcal{M}_d\}$ is at most $n+1$. In other words, $g$ has at most $n+1$ possible values when we range over all possible monotone partitions. Recall the definition of $q_{\min}(\delta) = \min_{(S_0, S_1) \notin B_\delta(S_0^\star, S_1^\star)} q(S_0, S_1)$. By the previous analysis and the Union Bound,

$$\mathbb{P}\left((S_0, S_1) \notin B_\delta(S_0^\star, S_1^\star)\right) \leq (n+2) \exp\left(-\frac{(q_{\min}(\delta_1, \delta_2) - q(S_0^\star, S_1^\star))^2 n}{36}\right)$$

Therefore, with probability at least

$$1 - (n+2) \exp\left(-\frac{(q_{\min}(\delta_1, \delta_2) - q(S_0^\star, S_1^\star))^2 n}{36}\right),$$

it holds that $\|\hat{f}_n - f\|_2^2 \leq \delta$.

For $d \geq 2$ and $(S_0, S_1) \in \mathcal{M}_d$, let

$$g\left(X_{1:n}, W_{1:n}; (S_0, S_1)\right) = \sum_{i=1}^n \mathbb{1}\left\{f(X_i + W_i) = 1, X_i \in S_0\right\} + \mathbb{1}\left\{f(X_i + W_i) = 0, X_i \in S_1\right\}.$$

The function $g$ represents the error associated with partition $(S_0, S_1)$. Applying the McDiarmid inequality,

$$\mathbb{P}\left(g\left(X_{1:n}, W_{1:n}; (S_0, S_1)\right) - \mathbb{E}\left[g\left(X_{1:n}, W_{1:n}; (S_0, S_1)\right)\right] \geq \epsilon n\right) \leq \exp\left(-\frac{\epsilon^2 n}{4}\right)$$

and

$$\mathbb{P}\left(g\left(X_{1:n}, W_{1:n}; (S_0, S_1)\right) - \mathbb{E}\left[g\left(X_{1:n}, W_{1:n}; (S_0, S_1)\right)\right] \leq -\epsilon n\right) \leq \exp\left(-\frac{\epsilon^2 n}{4}\right)$$

Calculating the expectation,

$$\begin{aligned}
\mathbb{E}\left[g\left(X_{1:n}, W_{1:n}; (S_0, S_1)\right)\right] &= n\left[p\int_{t \in S_1} h_0(t)dt + (1-p)\int_{t \in S_0} h_1(t)dt\right] \\
&= n\left[pH_0(S_1) + (1-p)H_1(S_0)\right] \\
&= n \cdot q(S_0, S_1).
\end{aligned}$$

By Assumption 2, the function $q(S_0, S_1)$ has a unique minimizer, $(S_0^\star, S_1^\star)$, that corresponds to the true function $f$. Therefore, if $\|\hat{f}_n - f\|_2^2$ is greater than $\delta$, then the function $\hat{f}_n$ must be outside of $B_\delta(S_0^\star, S_1^\star)$. Then it must be the case that some $(S_0, S_1)$ outside of $B_\delta(S_0^\star, S_1^\star)$ attained a lower value of $g$ than $g\left(X_{1:n}, W_{1:n}; (S_0^\star, S_1^\star)\right)$. We use concentration to upper bound the probability of this event.

First, we need to know how many possible objective values there are. This is upper bounded by the number of binary labelings of the set $\{X_1, \ldots, X_n\}$. By Theorem 6, it holds that

$$\mathbb{E}[L(X_1, \ldots X_n)] \leq \exp\left[(2^s + 2\log(2) - 1)\, n^{\frac{s-1}{s}}\right].$$

For any $\epsilon > 0$, the Markov inequality tells us that

$$\begin{aligned}
\mathbb{P}\left(L(X_1, \ldots X_n) \geq t\right) &\leq \frac{\mathbb{E}[L(X_1, \ldots X_n)]}{t} \\
&\leq \frac{\exp\left[(2^s + 2\log(2) - 1)\, n^{\frac{s-1}{s}}\right]}{t}.
\end{aligned}$$

Setting $t = \exp\left[n^{\frac{2s-1}{2s}}\right]$,

$$\mathbb{P}\left(L(X_1, \ldots X_n) \geq \exp\left[n^{\frac{2s-1}{2s}}\right]\right) \leq \frac{\exp\left[(2^s + 2\log(2) - 1)\, n^{\frac{s-1}{s}}\right]}{\exp\left[n^{\frac{2s-1}{2s}}\right]}.$$

Therefore, with probability at least $1 - \dfrac{\exp\left[(2^s + 2\log(2) - 1)n^{\frac{s-1}{s}}\right]}{\exp\left[n^{\frac{2s-1}{2s}}\right]}$, there are at most $\exp\left[n^{\frac{2s-1}{2s}}\right]$ label-

ings, and therefore function values. We bound the $L_2$ loss similarly to the proof for the case $d = s = 1$, above. Recall that $q_{\min}(\delta) = \min_{(S_0, S_1) \notin B_\delta(S_0^\star, S_1^\star)} q(S_0, S_1)$. Set $\epsilon = \frac{1}{3}(q_{\min}(\delta) - q(S_0^\star, S_1^\star))$ in the McDiarmid bound so that the optimal value remains separated from the alternatives.

$$
\begin{aligned}
&\mathbb{P}\left(\|\hat{f} - f\|_2^2 > \delta\right) \\
&= \mathbb{P}\left((S_0, S_1) \notin (B_\delta(S_0^\star, S_1^\star))\right) \\
&\leq \frac{\exp\left[(2^s + 2\log(2) - 1)\, n^{\frac{s-1}{s}}\right]}{\exp\left[n^{\frac{2s-1}{2s}}\right]} + \left(\exp\left[n^{\frac{2s-1}{2s}}\right] + 1\right)\exp\left(-\frac{(q_{\min}(\delta) - q(S_0^\star, S_1^\star))^2\, n}{36}\right)
\end{aligned}
$$

$\square$

*Proof of Corollary 4.* We equivalently show that $s = o\left(\sqrt{\log(n)}\right)$ is sufficient. Analyzing the first term,

$$
\begin{aligned}
\exp\left\{n\left(n^{s\log_n(2)} + 2\log(2) - 1 - n^{\frac{1}{2s}}\right)n^{-\frac{1}{s}}\right\} &\leq \exp\left\{n^{1-\frac{1}{s}}\left(n^{s\log_n(2)} - n^{\frac{1}{2s}} + \frac{1}{2}\right)\right\} \\
&= \exp\left\{n^{1-\frac{1}{2s}}\left(n^{s\log_n(2) - \frac{1}{2s}} - 1 + \frac{1}{2}n^{-\frac{1}{2s}}\right)\right\} \\
&\leq \exp\left\{n\left(n^{s\log_n(2) - \frac{1}{2s}} - \frac{1}{2}\right)\right\} \\
&= \exp\left\{n\left(n^{\frac{1}{s}\left(s^2\log_n(2) - \frac{1}{2}\right)} - \frac{1}{2}\right)\right\} \\
&= \exp\left\{n\left(n^{\frac{1}{s}\left(o(1) - \frac{1}{2}\right)} - \frac{1}{2}\right)\right\} \\
&= \exp\left\{n\left(n^{-\Theta(1)\frac{1}{s}} - \frac{1}{2}\right)\right\} \\
&= \exp\left\{n\left(n^{-\omega\left(\frac{1}{\sqrt{\log(n)}}\right)} - \frac{1}{2}\right)\right\} \\
&= \exp\left\{n\left(o\left(n^{-\frac{1}{\sqrt{\log(n)}}}\right) - \frac{1}{2}\right)\right\} \\
&= \exp\left\{n\left(o\left(n^{-\frac{1}{\log(n)}}\right) - \frac{1}{2}\right)\right\} \\
&= \exp\left\{n\left(o\left(n^{-\frac{\log_n(2)}{\log(2)}}\right) - \frac{1}{2}\right)\right\} \\
&= \exp\left\{n\left(o\left(2^{-\frac{1}{\log(2)}}\right) - \frac{1}{2}\right)\right\} \\
&= \exp\left\{n\left(o(1) - \frac{1}{2}\right)\right\} \\
&= \exp\left\{-\Theta(1)n\right\}
\end{aligned}
$$

We have assumed that the expression $(q_{\min}(\delta) - q(S_0^\star, S_1^\star))^2$ is constant in $s$. Analyzing the second term,

$$
\begin{aligned}
\exp\left[n^{\frac{2s-1}{2s}}\right]\exp\left(-\frac{(q_{\min}(\delta) - q(S_0^\star, S_1^\star))^2\, n}{36}\right) &= \exp\left\{n\left(n^{-\frac{1}{2s}} - \Theta(1)\right)\right\} \\
&= \exp\left\{n\left(n^{-\frac{1}{2o\left(\sqrt{\log(n)}\right)}} - \Theta(1)\right)\right\}
\end{aligned}
$$

$$= \exp \left\{ n \left( o \left( n^{-\frac{1}{2\sqrt{\log(n)}}} \right) - \Theta(1) \right) \right\}$$
$$= \exp \left\{ n \left( o(1) - \Theta(1) \right) \right\}$$
$$= \exp \left\{ -\Theta(1)n \right\}$$

$\square$

*Proof of Theorem 4.* The proof is analogous to the proof of Theorem 3, with the above definition for the function $q$. Recall that in the proof of Theorem 3, we needed to upper bound the number of possible function values. Here, the number of possible function values is upper bounded by the number of $s$-sparse binary labelings, which are those labelings corresponding to $s$-sparse monotone partitions. Let $L_s(X_1, \ldots X_n)$ be the number of $s$-sparse binary labelings.

By Theorem 6, it holds that

$$\mathbb{E}[L_s(X_1, \ldots X_n)] \leq \binom{d}{s} \exp \left[ (2^s + 2\log(2) - 1) \, n^{\frac{s-1}{s}} \right].$$

For any $\epsilon > 0$, the Markov inequality tells us that

$$\mathbb{P}\left( L_s(X_1, \ldots X_n) \geq t \right) \leq \frac{\mathbb{E}[L_s(X_1, \ldots X_n)]}{t}$$
$$\leq \frac{\binom{d}{s} \exp \left[ (2^s + 2\log(2) - 1) \, n^{\frac{s-1}{s}} \right]}{t}.$$

Setting $t = \binom{d}{s} \exp \left[ n^{\frac{2s-1}{2s}} \right]$,

$$\mathbb{P}\left( L_s(X_1, \ldots X_n) \geq \binom{d}{s} \exp \left[ n^{\frac{2s-1}{2s}} \right] \right) \leq \frac{\exp \left[ (2^s + 2\log(2) - 1) \, n^{\frac{s-1}{s}} \right]}{\exp \left[ n^{\frac{2s-1}{2s}} \right]}.$$

Therefore, with probability at least $1 - \frac{\exp \left[ (2^s + 2\log(2) - 1) n^{\frac{s-1}{s}} \right]}{\exp \left[ n^{\frac{2s-1}{2s}} \right]}$, there are at most $\binom{d}{s} \exp \left[ n^{\frac{2s-1}{2s}} \right]$ $s$-sparse binary labelings, and therefore function values. $\square$

*Proof of Corollary 5.* We have assumed that $s$ is constant and the sequence of functions $\{f_d\}$ extends a function of $s$ variables. For fixed $(S_0, S_1)$, the value of $q(S_0, S_1)$ does not change if we increase the overall dimension, because of the uniformity of $X$ and the independence of the coordinates of $W$. Therefore, $q$ does not depend on $d$ when $s$ is fixed, and so $q_{\min}(\delta) - q(S_0^\star, S_1^\star) = \Theta(1)$. We now analyze the bound in Theorem 4. Since $s$ is constant, the first term goes to zero. Analyzing the second term,

$$\left( \binom{d}{s} \exp \left[ n^{\frac{2s-1}{2s}} \right] + 1 \right) \exp \left( -\frac{(q_{\min}(\delta) - q(S_0^\star, S_1^\star))^2 \, n}{36} \right)$$
$$\leq \left( \exp \left[ s\log(d) + n^{\frac{2s-1}{2s}} \right] + 1 \right) \exp \left( -\frac{(q_{\min}(\delta) - q(S_0^\star, S_1^\star))^2 \, n}{36} \right)$$
$$= \exp \left[ s\log(d) + n^{\frac{2s-1}{2s}} - \Theta(1)n \right] + e^{-\Theta(1)n}$$

If $n = \omega(\log(d))$, the second term goes to zero. $\square$

The proof of Theorem 5 requires Lemmas 3 and 8.

*Proof of Lemma 3.* We need to show that

$$\mathbb{P}\left( Y_1 = 1, Y_2 = 0 | X_{1,k} > X_{2,k} \right) > \mathbb{P}\left( Y_1 = 0, Y_2 = 1 | X_{1,k} > X_{2,k} \right).$$

The proof is similar to the proof of Lemma 2. Consider the following procedure. We sample $X_1$ and $X_2$ independently and uniformly on $[0, 1]^d$. Fix $k \in A$. Let

$$X_+ = \begin{cases} X_1 & \text{if } X_{1,k} > X_{2,k} \\ X_2 & \text{otherwise} \end{cases}$$

and

$$X_- = \begin{cases} X_1 & \text{if } X_{1,k} \leq X_{2,k} \\ X_2 & \text{otherwise.} \end{cases}$$

In other words, $X_+$ is the right point according to coordinate $k$ and $X_-$ is the left point. As in the proof of Lemma 2, we can equivalently define $\bar{p}_k$ as

$$\bar{p}_k = \mathbb{P}\left(f(X_+ + W_1) > f(X_- + W_2)\right) - \mathbb{P}\left(f(X_+ + W_1) < f(X_- + W_2)\right).$$

Let $k = 1$. By Assumption 1, the function $f$ is not constant with respect to the first coordinate. Our goal is to show that

$$\mathbb{P}\left(f(X_+ + W_1) > f(X_- + W_2)\right) > \mathbb{P}\left(f(X_+ + W_1) < f(X_- + W_2)\right).$$

We now construct a coupling $(\overline{X}_+, \overline{X}_-, \overline{W}_1, \overline{W}_2) \sim (X_+, X_-, W_1, W_2)$ $(\star)$. Let $\overline{X}_{+,1} = X_{+,1}$ and $\overline{X}_{-,1} = X_{-,1}$. Let $\overline{X}_{+,i} = X_{-,i}$ and $\overline{X}_{-,i} = X_{+,i}$ for $i \in \{2, \ldots, d\}$. Finally, let $\overline{W}_1 = W_2$ and $\overline{W}_2 = W_1$. By monotonicity, $f(\overline{X}_+ + \overline{W}_1) \geq f(X_- + W_2)$. Similarly, $f(\overline{X}_- + \overline{W}_2) \leq f(X_+ + W_1)$. Therefore, the event $\{f(X_+ + W_1) < f(X_- + W_2)\}$ implies the event $\{f(\overline{X}_+ + \overline{W}_1) > f(\overline{X}_- + \overline{W}_2)\}$. Furthermore, $(\star)$ holds. This shows

$$\mathbb{P}\left(f(X_+ + W_1) > f(X_- + W_2)\right) \geq \mathbb{P}\left(f(X_+ + W_1) < f(X_- + W_2)\right).$$

To show a strict inequality, we need to show that the following event happens with positive probability.

$$\{f(\overline{X}_+ + \overline{W}_1) > f(\overline{X}_- + \overline{W}_2)\} \cap \{f(X_+ + W_1) \geq f(X_- + W_2)\}$$
$$= \{f(\overline{X}_+ + W_2) > f(\overline{X}_- + W_1)\} \cap \{f(X_+ + W_1) \geq f(X_- + W_2)\}$$

Observe that there exists $\epsilon$ such that $f(\overline{X}_+ + W_2) \geq f(X_- + W_2) + \epsilon$ and $f(\overline{X}_- + W_1) \leq f(X_+ + W_1) - \epsilon$ with positive probability. Otherwise, $f$ would be constant with respect to the first coordinate. This completes the proof. $\square$

The following proposition is the analogue of Proposition 6.

**Proposition 7.** *Consider stage $t$ in Algorithm 3'. Suppose that the first $t - 1$ coordinates recovered by the algorithm are correct, i.e. $k_i \in \{1, \ldots, s\}$ for all $i \in \{1, \ldots, t - 1\}$. Let $R = \{1, \ldots, s\} \setminus \{k_1, \ldots, k_{t-1}\}$. Let $(X_1, Y_1)$ and $(X_2, Y_2)$ be independent samples from the model. There exists $r \in R$ so that for all $k \in R$,*

$$\mathbb{P}\left(Y_1 > Y_2 | q(1, 2, r) = 1, q(1, 2, k) = 0\right) \geq \mathbb{P}\left(Y_2 > Y_1 | q(1, 2, r) = 1, q(1, 2, k) = 0\right).$$

*Proof.* The proof is identical to the proof of Proposition 6. $\square$

The following lemma is the analogue of Lemma 7.

**Lemma 8.** *Consider stage $t$ in Algorithm 3', which uses $m = \frac{n}{s}$ samples. The probability that the first coordinate is correctly recovered is at least*

$$1 - (d - 1) \exp\left(-\frac{p_1^2 m}{64 s^2}\right).$$

*Suppose that the first $t - 1$ coordinates recovered by the algorithm are correct, i.e. $k_i \in \{1, \ldots, s\}$ for all $i \in \{1, \ldots, t - 1\}$. Then $k_t \in \{1, \ldots, s\}$ with probability at least*

$$1 - (d - t) \exp\left(-\frac{p_1^2 m}{64 (s - t + 1)^2}\right).$$

*Proof.* The proof is nearly identical to the proof of Lemma 7, with $p$ replaced by $\overline{p}$. Lemma 3 guarantees that $\overline{p}_k > 0$ for all $k \in A$, and Proposition 7 establishes the existence of a special coordinate $r$. The bounded differences analysis for the application of the McDiarmid inequality again shows that each $X_i$ or $W_i$ can change the summation $\sum_{i=1}^{m} \sum_{j=1}^{m} \mathbb{1}\{Y_i > Y_j\}(q(i,j,r) - q(i,j,k))$ by up to $2(m-1)$. $\qquad\square$

*Proof of Theorem 5.* The proof is nearly identical to the proof of Theorem 2, and relies on Lemma 8. $\qquad\square$

*Proof of Corollary 6.* The proof is identical to the proof of Corollary 2, with $p$ replaced by $\overline{p}$. $\qquad\square$

*Proof of Corollary 7.* Support recovery fails with probability at most

$$ds \exp\left(-\frac{\overline{p}_1^2 n}{64 s^3}\right).$$

If it succeeds, the probability of the $L_2$ norm error exceeding $\delta$ is upper bounded by the value in Theorem 3. Then $\mathbb{P}\left(\|\hat{f}_n - f\|_2^2 > \epsilon\right)$ is at most

$$ds \exp\left(-\frac{\overline{p}_1^2 n}{64 s^3}\right) + \frac{\exp\left[(2^s + 2\log(2) - 1) m^{\frac{s-1}{s}}\right]}{\exp\left[m^{\frac{s-1}{s}+\epsilon}\right]} + \left(\exp\left[m^{\frac{s-1}{s}+\epsilon}\right] + 1\right) \exp\left(-\frac{(q_{\min}(\delta) - q(S_0^\star, S_1^\star))^2 m}{36}\right).$$

Therefore, if $n = \omega(s^3 \log(d))$ and $n = se^{\omega(s^2)}$, the estimator is consistent under the assumptions of Corollary 7. $\qquad\square$