[Reviews · NeurIPS 2019]

Reviewer 1



This paper has the feature of being one of the first (arguably the first) to address the problem of sparse isotonic regression, thus it is not easy to establish comparisons with other works. Moreover, isotonic regression is somewhat a niche area, although important, thus it is not clear what type of impact and significance this work may have. The paper is quite well written and, as far as I can assess (not having checked the proofs in detail) it is technically solid. In addition to a simple synthetic problem, the experimental section of the manuscript is focused on a single real-data experiment with gene expression data for cancer classification. One thing that is missing is an explanation of the motivation for using isotonic regression for this problem. I think the authors did a good job in addressing the reviewers' comments, so I'm keeping my original score 8 ("A very good submission; a clear accept.").

Reviewer 2



The proposed problems by the authors is interesting. Essentially one is looking for dimensions according to which the target variable becomes monotonic. The writing of the paper requires attention, and it suffers from the conference format. Most of the algoritmic side of the paper is pushed in the appendix, and the explanation of the algorithm is barebone: - it is not clear whether the problem is NP-hard or is it tractable. - Algorithm 1 requires integer programming, do you do this in practice, or is the algorithm just the theoretical construct? - Ignoring the relaxation of v_k, the authors do not prove that Algorithm 2/3+4 actually solves the same thing as Algorithm 1. Specifically, the objective in Eq. 10 is not properly explained, and how it is linked to Eq. 5. - Line 343 in Appendix seems incorrect: The new objective should be sum (1 - F_i)^2 + sum F_i^2 which is not a linear objective, note that F_i may be real numbers even if target is binary. This seems to be problematic for Lemma 1. The experiments should be improved. Additional benchmark dataserts would add more credibility. Also, the chosen baselines should perform relatively badly for the chosen data: 1. Based on Figure 1, the decision boundary is not linear which puts SVM at disadvantage. 2. barebone k-nn with (probably) high-dimensional data suffers form curse of dimension. A better comparison would be a decision tree and k-nn with feature selection/dimension reduction.

Reviewer 3



Isotonic regression is an important model both in theory and practice, and thus I think the work in this article is meaningful. As suggesting in the theorem, it is desirable for estimator to have the property such as consistency and support recovery, however, I'm interested in the convergence rate is how optimal. Actually, the convergence rate would seem a little bit slow in practice. In experimental results, the authors check the performance of the support recovery, while estimation accuracy should also be assessed for confirming the consistency. In addition, for example in Theorem 2, $N=sn$ is defined as a samples for algorithm. However, since the input of algorithm is only $n$ sample and sparsity level, I'm not sure actually $N$ means. Please make it clear. As a minor comment, Algorithm 3 in Theorem 3 would be Algorithm 1 from the context.

[Author Response · NeurIPS 2019]

We thank the reviewers for their comments.

**Review #1**: Regarding the significance and impact of the work, isotonic regression has been used in a multitude of applications, a few of which are given in the introduction. One of the most natural application areas for isotonic regression is biology. Biologists have established that "genetic effects on phenotypes such as height, fitness or disease are monotone'" ([1]). See [1] for references to the biology literature, and [2] for a discussion on monotonic genetic effects on disease.

In the case of disease, the presence or absence of a disease follows a monotone relationship with respect to gene expression. Classifying between lung and skin cancer amounts to applying this principle to a subpopulation of individuals who have lung or skin cancer. We will certainly include this reasoning in the revision of our paper. The motivation for a *sparse* model is that certain genes should be more responsible for disease than others. Sparsity can be viewed as a kind of regularization; to prevent overfitting, we allow the regression to explain the results using only a small number of genes. By identifying the most relevant genes, sparse isotonic regression helps elucidate the mechanism of disease. We have discussed our work with a biostatistician who works on cancer detection, and are working on using applying our algorithms to histology data from his lab. We hope this will improve the detection accuracy of his method.

**Review #2**: We agree with your idea to move the presentation of the algorithms to the main text while moving the results in Section 4 to the Appendix. Regarding tractability, the sparse quadratic minimization problem solved by the IPIR algorithm is NP-hard; we will include a reference. Both LPSR and S-LPSR are linear programs, which can be solved in polynomial time. The second step of TSIR in the Noisy Output Model is a linearly constrained quadratic program that can be solved in polynomial time. TSIR in the Noisy Input Model can also be solved in polynomial time. A note about Lemma 1: we should have stated that for the Noisy Input Model, there is a polynomial-time procedure to obtain an optimal solution to Problem (21)-(23), by forming the associated linear program and then finding an integer optimal solution. The procedure to find an integer optimal solution is part of the proof of Lemma 1. We agree that it is important to discuss tractability of algorithms, so all of this will be clarified in the revision.

Regarding line 343 in the Appendix: Constraint (9) requires each $F_i$ to be either 0 or 1. Therefore, the substitution that we gave for the objective function gives an equivalent integer linear program.

Algorithm 1 is indeed implemented with integer programming in Gurobi, as written. The algorithm can be quite slow, which motivates the need for the two-stage approach. Algorithm 2/3+4 does not solve the same thing as Algorithm 1, but rather is a heuristic. The idea is that a problem becomes more tractable when decisions are made in two steps instead of simultaneously. To clarify, the objective in Eq. 10 is tailored to the goal of support recovery alone, and should be viewed separately from Eq. 5.

Regarding experiments, we have now implemented k-NN with dimension reduction at your suggestion. Surprisingly, the performance is worse than k-NN, achieving close to the baseline performance of about $68\%$. Even without comparison to other approaches, the excellent performance of our algorithm shows that it is a promising approach.

**Review #3**: Regarding the optimality of the convergence rate, it is true that the error bounds are probably quite loose, due to large constants appearing in the bounds. Our main goal was to show statistical consistency of our algorithms, and we did not optimize the constants. Tightening the bounds would be an interesting direction for future research. We note that error bounds are given for a noisy input setting, which is not typically seen in the literature but is often encountered in practice, such as the cancer application we studied.

Regarding the synthetic experiments, we have now run 50 trials. For the final version of the paper, many more trials will be conducted. In addition to the original metric measuring the frequency of the correct support recovery by our computations, per the reviewer's suggestion we will also report our results on function estimation accuracy. We found those to be extremely encouraging. For example, with 250 samples and d = 10, IPIR had a function estimation accuracy of 92.2%, LPSR had an accuracy of 89.2%, and S-LPSR had an accuracy of 90.1%. The accuracy was measured on 500 data samples.

To clarify the confusion about $n$ and $N$, Theorem 2 requires fresh samples to be used at each iteration of S-LPSR. Given $N$ samples, we divide into $s$ batches of size $n$, corresponding to the $s$ iteration steps. We apologize for the confusing presentation, and will clarify this in the revision.

# References

[1] Qiyang Han, Tengyao Wang, Sabyasachi Chatterjee, and Richard J. Samworth. Isotonic regression in general dimensions. *arXiv 1708.0946v1*, 2017.

[2] Ramamurthy Mani, Robert P. St. Onge, John L. Hartman IV, Guri Giaever, and Frederick P. Roth. Defining genetic interaction. *Proceedings of the National Academy of Sciences of the United States of America (PNAS)*,


[Meta-Review · NeurIPS 2019]

Congratulations! The reviewers mostly appreciated the paper, and have made several suggestions for improving the presentation and clarity (in their original reviews). We hope that you will incorporate these (and the other changes discussed in your rebuttal) in the final version.